# Robust Adversarial Reinforcement Learning via Bounded Rationality Curricula

**Aryaman Reddi**[1,2*]**, Maximilian Tölle**[1,3]**, Jan Peters**[1,2,3,4]**,**
**Georgia Chalvatzaki**[1,2]**, Carlo D'Eramo**[1,2,5]

[1]Department of Computer Science, TU Darmstadt, Germany
[2]Hessian Center for Artificial Intelligence (Hessian.ai), Germany
[3]German Research Center for AI (DFKI), Systems AI for Robot Learning, Germany
[4]Center for Cognitive Science, TU Darmstadt, Germany
[5]Center for Artificial Intelligence and Data Science, University of Würzburg, Germany

## Abstract

Robustness against adversarial attacks and distribution shifts is a long-standing goal of Reinforcement Learning (RL). To this end, Robust Adversarial Reinforcement Learning (RARL) trains a protagonist against destabilizing forces exercised by an adversary in a competitive zero-sum Markov game, whose optimal solution, i.e., *rational strategy*, corresponds to a Nash equilibrium. However, finding Nash equilibria requires facing complex saddle point optimization problems, which can be prohibitive to solve, especially for high-dimensional control. In this paper, we propose a novel approach for adversarial RL based on entropy regularization to ease the complexity of the saddle point optimization problem. We show that the solution of this entropy-regularized problem corresponds to a Quantal Response Equilibrium (QRE), a generalization of Nash equilibria that accounts for bounded rationality, i.e., agents sometimes play random actions instead of optimal ones. Crucially, the connection between the entropy-regularized objective and QRE enables free modulation of the rationality of the agents by simply tuning the temperature coefficient. We leverage this insight to propose our novel algorithm, Quantal Adversarial RL (QARL), which gradually increases the rationality of the adversary in a curriculum fashion until it is fully rational, easing the complexity of the optimization problem while retaining robustness. We provide extensive evidence of QARL outperforming RARL and recent baselines across several MuJoCo locomotion and navigation problems in overall performance and robustness.

## 1 Introduction

The glaring success of deep reinforcement learning (RL) has been largely obtained in controlled simulated problems under no or negligible disturbances (Haarnoja et al., 2018a; Mnih et al., 2015; Barth-Maron et al., 2018; Schulman et al., 2017; Fujimoto et al., 2018). However, deep RL methods are widely known to be prone to overfit the observed tasks, such that distribution shifts, or even adversarial attacks, can dramatically undermine their performance (Gleave et al., 2019; Zhang et al., 2018). For the ultimate deployment of deep RL in realistic high-dimensional problems, it is of pivotal importance to endow agents with *robustness* w.r.t. unmodeled perturbations. Robust Adversarial Reinforcement Learning (RARL) is an appealing approach to obtain robust policies in RL by training a protagonist agent against destabilizing actions applied by an adversarial agent (Pinto et al., 2017). RARL solves a two-player zero-sum Markov game (Littman, 1994), where the protagonist can execute actions in the environment to maximize a measure of performance, and the adversary is trained to perform adversarial actions to minimize the same measure. By learning tasks under destabilizing perturbations, the protagonist obtains robust skills counteracting distribution shifts and adversarial attacks when deployed.

---

*Correspondence to `<aryaman.reddi@tu-darmstadt.de>`

The zero-sum Markov game solved by RARL is a saddle point optimization problem whose solution is the minimax or Nash equilibrium, which can be prohibitive to find in case of strongly non-convex non-concave objective functions typical when dealing with deep RL control problems (Ostrovskii et al., 2021). RARL tackles these issues by adopting an alternating approach where one agent is trained while the other is stationary. This iterative approach is inspired by the fictitious-play heuristic of game theory (Brown, 1951; Heinrich & Silver, 2016) that, despite not ensuring convergence, performs well in practice (Hofbauer & Sandholm, 2002; Brandt et al., 2010). Nevertheless, several works have shown that RARL can fail to converge to good solutions, being unstable even in basic linear quadratic regulation (Zhang et al., 2020a). Attempts have been made to tackle the tendency of RARL to stick to suboptimal solutions, e.g., perturbing the gradient with Langevin dynamics (Kamalaruban et al., 2020) or using extragradient optimization (Cen et al., 2021).

In this paper, we propose a new approach based on entropy regularization to ease the complexity of the adversarial RL optimization problem. We formulate an entropy-regularized zero-sum Markov game where two agents can compete while maximizing the entropy of their respective policies. We show that the solution to this problem is a Quantal Response Equilibrium (QRE), a generalization of the Nash equilibrium that accounts for *bounded rationality* of the agents (Samuelson, 1995; McKelvey & Palfrey, 1995; Goeree et al., 2016). We note that, in a setting where the protagonist and the adversary are completely rational, i.e., they execute their policies without noise, the solution is a saddle point corresponding to a Nash equilibrium. This solution ensures the most robust behavior (Pinto et al., 2017), but it is typically difficult to find due to the strongly non-convex non-concave optimization problem (Kamalaruban et al., 2020; Zhang et al., 2020a). Conversely, in a setting where the protagonist is completely rational and the rationality of the adversary is minimal, i.e., the adversary plays only random actions, the optimization problem reduces to a regular maximization problem for the protagonist at the cost of lower robustness of the obtained policy. This entails that there is a trade-off between the complexity of the optimization problem and the robustness of the obtained policy, and we posit that an effective balance can be found by properly tuning the rationality of the adversary.

It can be shown that the *temperature* parameter of the entropy-regularized problem corresponds to the rationality parameter of the respective QRE (Savas et al., 2019; Cen et al., 2021), which affects the complexity of the optimization problem and the robustness of the optimal solution. Based on this, our main contribution is a novel algorithm for adversarial RL, that we call Quantal Adversarial RL (QARL), that *actively modulates the rationality of the adversary to ease the complexity of the optimization problem while retaining the robustness of the obtained solution*. QARL initially solves an adversarial problem with a completely random adversary, thus solely focusing on the performance improvement of the protagonist, neglecting robustness. Then, it gradually increases the rationality of the adversary until it reaches complete rationality and, hence, maximal robustness. While an arbitrary method can be chosen for changing the rationality of the adversary, we provide a curriculum-like approach that automatically tunes it while keeping track of the learning progress of the protagonist. This way, we can maintain a steady performance improvement together with the increase in robustness. We demonstrate that our approach facilitates the learning of the protagonist, and we show that QARL outperforms RARL and related baselines in terms of performance and robustness across several high-dimensional MuJoCo problems, such as navigation and locomotion, of the DeepMind Control Suite (Todorov et al., 2012; Tunyasuvunakool et al., 2020).

## 2 RELATED WORKS

Over the last decades, unwavering effort has been put into endowing learning agents with robust behavior in RL (Moos et al., 2022). Some recent attempts are based on a zero-sum game problem formulation, also known as robust adversarial RL (RARL), where an adversary agent is trained to exert destabilizing actions on a protagonist agent trying to accomplish a given task (Pinto et al., 2017; Oikarinen et al., 2021). This way, the protagonist learns to solve a task while counteracting external disturbances, thus being more robust when deployed. The quest initiated by RARL for achieving robustness with adversarial RL has been pursued by several subsequent works that attempt to overcome the limitations of RARL. For example, Pan et al. (2019) proposes to model risk explicitly to avoid catastrophic behavior, while Zhang et al. (2020a) shows that RARL is affected by instability and convergence issues even in simple linear quadratic regulation problems. Furthermore, Kamalaruban et al. (2020) proposes the use of Langevin dynamics for optimizing the challenging non-convex

non-concave objective of the zero-sum game of RARL, while Zhai et al. (2022) studies the benefit of imposing dissipation-inequation-constraints on the adversary. Similarly to this paper, some works investigate the potential of curricula in adversarial RL, e.g., motivated by $H_\infty$-control, they build a sequence of increasingly complex subtasks (Song & Schneider, 2022; Ao et al., 2022). Sheng et al. (2022) proposes to generate curricula over the strength of an adversary that exerts adversarial attacks on the protagonist by acting on its action space. Of these, we consider Kamalaruban et al. (2020) and Sheng et al. (2022), as they both aim to improve the robustness and stability of the robust adversarial RL scheme, which is the primary aim of this work. Pan et al. (2019) aims to avoid catastrophic policy collapse, while Zhai et al. (2022) uses model-based disturbance bounding which requires more system knowledge than we assume. Song & Schneider (2022) and Ao et al. (2022) encourage task robustness through adversarial subtask generation as opposed to adversarial disturbance modeling, which goes beyond the scope of the non-hierarchical perspective we adopt, but could nonetheless be interesting for future work.

## 3    PRELIMINARIES

We consider two-player zero-sum Markov games (Littman, 1994; Perolat et al., 2015) formulated as a Markov decision process (Puterman, 1990) $\mathcal{M} = \langle \mathcal{S}, \mathcal{A}_1, \mathcal{A}_2, \mathcal{R}, \mathcal{P}, \gamma, \iota \rangle$, where $\mathcal{S}$ is the continuous state space observed by both players, $\mathcal{A}_1$ and $\mathcal{A}_2$ are the continuous action spaces of the two agents, and $\mathcal{P} : \mathcal{S} \times \mathcal{A}_1 \times \mathcal{A}_2 \times \mathcal{S} \to \mathbb{R}$ is a transition probability density, $\mathcal{R} : \mathcal{S} \times \mathcal{A}_1 \times \mathcal{A}_2 \times \mathcal{S} \to \mathbb{R}$ is a reward function, $\gamma \in [0, 1)$ is a discount factor, and $\iota$ a probability density function over initial states. For given policies $\mu$ and $\nu$ of the two players, the discounted return is defined as

$$J_{\mu,\nu}(s) = \mathbb{E}_{a^1 \sim \mu(\cdot|s), a^2 \sim \nu(\cdot|s)} \left[ \sum_{t=0}^{H-1} \gamma^t \mathcal{R}(s_t, a_t^1, a_t^2, s_{t+1}) \Big| s_0 = s, \mathcal{P} \right] , \forall s \in \mathcal{S}. \tag{1}$$

In a two-player zero-sum Markov game, one agent has to obtain a policy $\mu$ that maximizes the discounted return $J$, while the other aims at obtaining a policy $\nu$ that minimizes it. The optimal policies $\mu^*$ and $\nu^*$ are the ones inducing the return

$$J_{\mu^*,\nu^*} = \max_\mu \min_\nu J_{\mu,\nu} = \min_\nu \max_\mu J_{\mu,\nu}, \tag{2}$$

which corresponds to a minimax or Nash equilibrium for the Markov game (Perolat et al., 2015). It is well-known that obtaining Nash equilibria in Markov games is a challenging problem that requires solving minimax equilibria at every state (Littman, 1994; Pérolat et al., 2017; Song et al., 2023). Most methods to compute optimal policies suffer from exponential complexity in the number of actions for discrete Markov games; hence, they are impractical for problems with many actions and unfeasible for high-dimensional control problems.

**Quantal response equilibrium and bounded rationality**   Finding a Nash equilibrium requires solving a saddle point optimization problem, for which the corresponding knife-edge best-response function can be challenging to optimize due to strong non-convexity non-concavity. The Quantal Response Equilibrium (QRE) is a generalization of the Nash equilibrium, which can arbitrarily smoothen best-response functions, thus easing the optimization (McKelvey & Palfrey, 1995; Goeree et al., 2016). In practice, smoothing best-response functions results in modeling games where players do not always select their best strategy, sometimes selecting suboptimal actions. This behavior is known as *bounded rationality*, in contrast to full rationality typical of knife-edge best-response strategies, i.e., Nash equilibria. Consider a two-agent Markov game with a given payoff $X$, a parameter $\tau \in \mathbb{R}^+$, and two available actions per agent. Denote the strategies as $\sigma_{ij}$ where $i$ and $j$ indicate the agent and the action, respectively. A particular form of QRE, known as *logit QRE* and used across our work, is

$$\sigma_{11}^* = \frac{\exp(X\sigma_{21}^*/\tau)}{\exp(X\sigma_{21}^*/\tau) + \exp(X\sigma_{22}^*/\tau)} \quad , \quad \sigma_{21}^* = \frac{\exp(X\sigma_{12}^*/\tau)}{\exp(X\sigma_{11}^*/\tau) + \exp(X\sigma_{12}^*/\tau)}, \tag{3}$$

with $\sigma_{12}^* = 1 - \sigma_{11}^*$ and $\sigma_{22}^* = 1 - \sigma_{21}^*$. By adopting a logistic function shape, the logit QRE models strategies with bounded rationality depending on the expected payoff $X$ and a belief $\sigma^*$ about the strategy of the other agent, where $\tau$ modulates the rationality level of the agents. For $\tau \to \infty$, the agents play completely random actions; on the contrary, when $\tau \to 0$, the strategies become completely rational, thus becoming a Nash equilibrium.

**Robust adversarial reinforcement learning**  A special case of a two-player zero-sum Markov game is one where an adversary agent is trained to execute destabilizing actions on a protagonist trained to accomplish a given task (Kamalaruban et al., 2020; Pinto et al., 2017; Zhang et al., 2020a). By learning to deal with external disturbances while solving its task, the protagonist develops skills robust to adversarial attacks and distribution shifts when deployed. RARL (Pinto et al., 2017) is an algorithmic solution for solving such two-player Markov games that obtains robust behavior in high-dimensional control problems, e.g., MuJoCo (Todorov et al., 2012). RARL consists of a loop of separate training phases for the protagonist and the adversary, i.e., the protagonist performs rollouts and training steps while the adversary is stationary and vice versa. At each timestep $t$, both agents observe the same state $s_t \in \mathcal{S}$, perform actions $a_t^1 \in \mathcal{A}_1$ and $a_t^2 \in \mathcal{A}_2$, and reach state $s_{t+1}$, while the protagonist obtains reward $r_t^1 = \mathcal{R}(s_t, a_t^1, a_t^2, s_{t+1})$ and the adversary gets $r_t^2 = -r_t^1$.

## 4    BOUNDED RATIONALITY IN ADVERSARIAL REINFORCEMENT LEARNING

A prerequisite for our approach is the ability to model and control the rationality of agents in an adversarial RL setting. Crucially, entropy regularization provides a theoretical framework to model Markov games under bounded rationality (Cen et al., 2021; Savas et al., 2019). Consider the following maximum entropy formulation of the adversarial RL objective (2)

$$J_{\mu^*,\nu^*} = \max_{\mu} \min_{\nu} J_{\mu,\nu} + \beta \mathcal{H}(\mu) - \alpha \mathcal{H}(\nu), \tag{4}$$

where $\mathcal{H}$ is the Shannon entropy of a distribution and $\alpha, \beta \in \mathbb{R}^+$ are two temperature coefficients. The solution to this problem is known as Quantal Response Equilibrium (QRE) (McKelvey & Palfrey, 1995; Goeree et al., 2016), a generalization of the Nash equilibrium that extends it to games where agents do not play with complete rationality[1].

**Definition 4.1** *For any entropy-regularized zero-sum Markov game, the QRE is*

$$\begin{cases} \mu^*(a^1|s) = \frac{\exp\left(J_{\nu^*}(s,a^1)/\beta\right)}{\int_b \exp\left(J_{\nu^*}(s,b)/\beta\right)} \propto \exp\left(J_{\nu^*}(s,a^1)/\beta\right), & \forall s \in \mathcal{S}, a^1 \in \mathcal{A}_1 \\[2mm] \nu^*(a^2|s) = \frac{\exp\left(-J_{\mu^*}(s,a^2)/\alpha\right)}{\int_b \exp\left(-J_{\mu^*}(s,b)/\alpha\right)} \propto \exp\left(-J_{\mu^*}(s,a^2)/\alpha\right), & \forall s \in \mathcal{S}, a^2 \in \mathcal{A}_2, \end{cases} \tag{5}$$

*where $J_\mu(s,a)$ (or $J_\nu(s,a)$) indicates the performance obtained when the adversary (or protagonist) executes action $a$ in state $s$ while the protagonist (or adversary) follows its policy $\mu$ (or $\nu$).*

The solution (5) to the entropy-regularized objective (4) can be considered as the adversarial RL counterpart of the well-known solution for maximum entropy RL (Haarnoja et al., 2018a; 2017; Asadi & Littman, 2017). Key to our analysis is that the temperature coefficients $\alpha$ and $\beta$ can effectively modulate the rationality of the agents under a game-theoretical lens (McKelvey & Palfrey, 1995; Goeree et al., 2016). For example, for $\alpha \to \infty$, the entropy of the adversary is maximized, and its rationality is minimized. Conversely, if $\alpha \to 0$, the entropy-regularized objective (4) accounts only for the performance term, and the adversary is fully rational.

### 4.1    TEMPERATURE-CONDITIONED VALUE FUNCTIONS AND POLICIES

QRE are shown to exist in finite Markov games with bounded rationality (McKelvey & Palfrey, 1995; Goeree et al., 2016). However, in Markov games with large state-action spaces, solving the QRE in closed-form becomes computationally unfeasible and RL becomes more suitable at solving the two-player optimisation. We point out that QRE closely resembles the solution of the maximum entropy RL problem. For example, the well-known soft-actor critic (SAC) algorithm (Haarnoja et al., 2018a;b) approximates policies of the form $\pi(\cdot|s) \approx \frac{\exp\left(Q(s,\cdot)/\tau\right)}{Z(s,\tau)}$, where $Z(\cdot)$ is a partition function and $\tau \in \mathbb{R}^+$ is the temperature. Thus, we introduce the use of SAC in adversarial RL to obtain a practical method for approximating QRE in the case of continuous actions.

For our purpose of controlling the rationality of the adversary, we propose to condition its policy on the temperature coefficient. To do this, we define what we call temperature-conditioned value

---

[1] QRE are homogeneous for $\alpha = \beta$. Here, we use *heterogeneous* QRE, i.e., $\alpha \neq \beta$ (Goeree et al., 2016; Rogers et al., 2009).

functions and policies, whose definition follows directly from the one for soft action-value function in Theorem 1 of Schaul et al. (2015) and Haarnoja et al. (2017).

**Corollary 4.2** *For any policy $\pi$ and temperature $\tau \in \mathbb{R}^+$, let the temperature-conditioned optimal soft action-value function and value function be defined as*

$$Q^*(s_t, a_t, \tau) = r_t + \mathbb{E}\left[\sum_{i=1}^{\infty} \gamma^i \left(r_{t+i} + \tau \mathcal{H}(\pi^*(\cdot|s_{t+i}, \tau)))\right)\Big|\mathcal{P}\right], \tag{6}$$

$$V^*(s_t, \tau) = \tau \log \int_{\mathcal{A}} \exp\left(\frac{Q^*(s_t, a', \tau)}{\tau}\right) da'. \tag{7}$$

*Then, the optimal policy conditioned on temperature $\tau$ is*

$$\pi^*(a_t|s_t, \tau) = \exp\left(\frac{Q^*(s_t, a_t, \tau) - V^*(s_t, \tau)}{\tau}\right). \tag{8}$$

The temperature-conditioned policy $\pi^*$ (Equation (8)) can be straightforwardly rewritten as $\pi^*(a_t|s_t, \tau) = \frac{\exp(Q^*(s_t, a_t, \tau)/\tau)}{\int_{\mathcal{A}} \exp(Q^*(s_t, a', \tau)/\tau) da'}$. Using the action-value function $Q^*$ as an estimate of $J_{\pi^*}$, the resemblance of $\pi^*$ with QRE (Equation (5)) is clear, which enables us to derive the following.

**Proposition 4.3** *For any entropy-regularized zero-sum Markov game $\mathcal{M}$ as defined in Equation (4), given the temperature-conditioned optimal action-value function $Q^*$ and value function $V^*$, the temperature-conditioned policies are*

$$\begin{cases} \mu^*(a^1|s, \beta) = \frac{\exp(Q^*_\nu(s, a^1, \beta)/\beta)}{\int_b \exp(Q^*_\nu(s, b, \beta)/\beta)} \propto \exp(Q^*_\nu(s, a^1, \beta)/\beta), & \forall s \in \mathcal{S}, a^1 \in \mathcal{A}_1 \\ \nu^*(a^2|s, \alpha) = \frac{\exp(-Q^*_\mu(s, a^2, \alpha)/\alpha)}{\int_b \exp(-Q^*_\mu(s, b, \alpha)/\alpha)} \propto \exp(-Q^*_\mu(s, a^2, \alpha)/\alpha), & \forall s \in \mathcal{S}, a^2 \in \mathcal{A}_2, \end{cases} \tag{9}$$

*where $Q^*_\pi(s, a, \tau)$ indicates the performance conditioned on temperature $\tau$ obtained when one agent executes action $a$ in state $s$ while the other agent follows its policy $\pi$. We have that $\langle \mu^*, \nu^* \rangle$ is a QRE for the Markov game $\mathcal{M}$.*

This result establishes a connection between QRE and optimal policies in two-player Markov games, which enables modeling adversarial RL problems with bounded rationality.

## 4.2 AUTOMATIC GENERATION OF BOUNDED RATIONALITY CURRICULA

So far, we have a generalization of adversarial RL to bounded rationality through entropy regularization and a simple way to tune rationality by acting on the temperature coefficient. Crucially, we are now able to shape the saddle point optimization problem of adversarial RL by modulating the rationality of the adversary through its temperature $\alpha$. We recall that our goal is to ease the complexity of potentially strongly non-convex non-concave optimization problems, typical in high-dimensional control while retaining the robustness of the obtained policies. We propose to initially solve an adversarial problem with a completely random adversary, i.e., $\alpha \to \infty$, which results in a simpler plain maximization of the performance of the protagonist, neglecting robustness. Then, we gradually increase the rationality of the agent and thus the complexity of the optimization problem by consciously decreasing the temperature $\alpha$, hence improving the robustness of the obtained policy. Eventually, we have $\alpha \to 0$, retrieving the regular adversarial RL problem with a completely rational adversary and, consequently, maximal robustness of the obtained policy. The benefit of our approach is that, instead of attempting to solve a complex saddle point optimization from the beginning, it starts from a simpler one and makes it gradually more complex in a curriculum fashion (Zhang et al., 2020b; Svetlik et al., 2017; Klink et al., 2020; 2021).

We are left with the decision of how to tune the temperature of the adversary during training. While a viable solution would be to gradually anneal the temperature during training, it is prohibitive to manually select a proper decay speed that ensures a good balance between the complexity of the problem and its feasibility. Ideally, the temperature should be gradually decreased to increase rationality while avoiding a negative impact on the performance of the protagonist. Therefore, we

---

**Algorithm 1** Quantal Adversarial Reinforcement Learning (QARL)

1: **Input:** Initial critic parameters $\psi_1^1, \psi_2^1, \bar{\psi}_1^1, \bar{\psi}_2^1, \psi_1^2, \psi_2^2, \bar{\psi}_1^2, \bar{\psi}_2^2$; initial actor parameters $\phi^1, \phi^2$; $\mathcal{D}^1 = \emptyset, \mathcal{D}^2 = \emptyset$; initial temperature distribution parameters $\boldsymbol{\omega}$; performance lower bound $\xi$; #sampled temperatures $N$; #Monte-Carlo rollouts $M$; initial temperatures $\alpha, \beta$; #steps $T$.
2: **for** each iteration **do**
3:      $\bar{\psi}_1^1 \leftarrow \psi_1^1, \bar{\psi}_2^1 \leftarrow \psi_2^1, \bar{\psi}_1^2 \leftarrow \psi_1^2, \bar{\psi}_2^2 \leftarrow \psi_2^2$             ▷ Target networks update
4:      $\alpha_{1,\ldots N} \sim p_{\boldsymbol{\omega}}(\alpha)$             ▷ Sampling of temperatures of the adversary
5:      **for** $i = 1, \ldots, N$ **do**
6:          **for** $t = 1, \ldots, T$ **do**             ▷ Transitions collection for the adversary
7:              $a_t^1 \sim \mu_{\phi^1}(a_t^1|s_t, \alpha_i)$
8:              $a_t^2 \sim \nu_{\phi^2}(a_t^2|s_t, \alpha_i)$
9:              $s_{t+1} \sim \mathcal{P}(s_{t+1}|s_t, a_t^1, a_t^2)$
10:              $\mathcal{D}^2 = \mathcal{D}^2 \cup \{\langle s_t, a_t^2, r_t, s_{t+1}, \alpha_i\rangle\}$
11:              $\psi_1^2, \psi_2^2, \phi^2 \leftarrow \texttt{ADVERSARY\_UPDATE}(\psi_1^2, \psi_2^2, \phi^2, \mathcal{D}^2)$
12:          **end for**
13:      **end for**
14:      **for** $i = 1, \ldots, N$ **do**
15:          **for** $t = 1, \ldots, T$ **do**             ▷ Transitions collection for the protagonist
16:              $a_t^1 \sim \mu_{\phi^1}(a_t^1|s_t, \alpha_i)$
17:              $a_t^2 \sim \nu_{\phi^2}(a_t^2|s_t, \alpha_i)$
18:              $s_{t+1} \sim \mathcal{P}(s_{t+1}|s_t, a_t^1, a_t^2)$
19:              $\mathcal{D}^1 = \mathcal{D}^1 \cup \{\langle s_t, a_t^1, r_t, s_{t+1}, \alpha_i\rangle\}$
20:              $\psi_1^1, \psi_2^1, \phi^1, \beta \leftarrow \texttt{PROTAGONIST\_UPDATE}(\psi_1^1, \psi_2^1, \phi^1, \beta, \mathcal{D}^1)$
21:          **end for**
22:      **end for**
23:      Update $\boldsymbol{\omega}$ optimizing (10), estimating $\mathbb{E}_{p_{\boldsymbol{\omega}}(\alpha)}[J_{\mu,\nu}(\boldsymbol{\omega})]$ with $M$ rollouts
24: **end for**
25: **Return:** Trained actor parameters $\phi^1, \phi^2$

---

provide a way that automatically tunes the temperature while keeping track of the learning progress of the protagonist. We define a probability distribution $p_{\boldsymbol{\omega}}(\alpha)$, parameterized by $\boldsymbol{\omega}$, over the temperature coefficient $\alpha$ of the adversary in Objective (4). Given that $\alpha \in \mathbb{R}^+$, it is convenient to model the probability distribution $p_{\boldsymbol{\omega}}$ as a gamma distribution $\Gamma(k, \theta)$, where $\boldsymbol{\omega} = \langle k, \theta \rangle$. The parameters of the distribution are gradually updated to reach a target distribution $\mu(\alpha)$, under the constraint that the update does not result in decreasing the performance $J_{\mu,\nu}$ below a certain threshold $\xi$. Since we want the adversary to converge to complete rationality (i.e., $\alpha \to 0$), as in RARL, we set the target distribution to a sharp gamma distribution $\mu = \Gamma(1, 10^{-3})$, approximating a Dirac distribution centered in $\alpha = 0$. Formally, we solve the following constrained optimization problem

$$\min_{\boldsymbol{\omega}} \ D_{\text{KL}}\left(p_{\boldsymbol{\omega}}(\alpha)||\mu(\alpha)\right)$$
$$\text{s.t. } \mathbb{E}_{p_{\boldsymbol{\omega}}(\alpha)}\left[J_{\mu,\nu}(\alpha)\right] \geq \xi$$
$$D_{\text{KL}}\left(p_{\boldsymbol{\omega}}(\alpha)||p_{\boldsymbol{\omega}'}(\alpha)\right) \leq \epsilon, \tag{10}$$

where $\boldsymbol{\omega}'$ is the parameter vector before the update. By minimizing the KL-divergence between the current distribution $p_{\boldsymbol{\omega}}$ and $\mu$, we progressively sample lower $\alpha$, thus increasing the rationality of the adversary. While the second constraint avoids abrupt changes in the distribution $p_{\boldsymbol{\omega}}$, the first constraint crucially ensures that the update of the distribution $p_{\boldsymbol{\omega}}$ does not make the overall performance drop below a certain threshold $\xi$. This way, QARL progressively increases the rationality of the adversary while preventing catastrophic performance drops, eventually reaching an adversary with full rationality, and thus maximal robustness, as in RARL (Pinto et al., 2017).

### 4.3 QUANTAL ADVERSARIAL REINFORCEMENT LEARNING

The constrained optimization problem (10) can be solved by minimizing its Lagrangian

$$\mathcal{L}(\boldsymbol{\omega}, \lambda, \eta) = D_{\text{KL}}\left(p_{\boldsymbol{\omega}}(\alpha)||\mu(\alpha)\right) + \lambda\left(\xi - \mathbb{E}_{p_{\boldsymbol{\omega}}(\alpha)}\left[J_{\mu,\nu}(\boldsymbol{\omega})\right]\right) + \eta\left(D_{\text{KL}}\left(p_{\boldsymbol{\omega}}(\alpha)||p_{\boldsymbol{\omega}'}(\alpha)\right) - \epsilon\right), \tag{11}$$

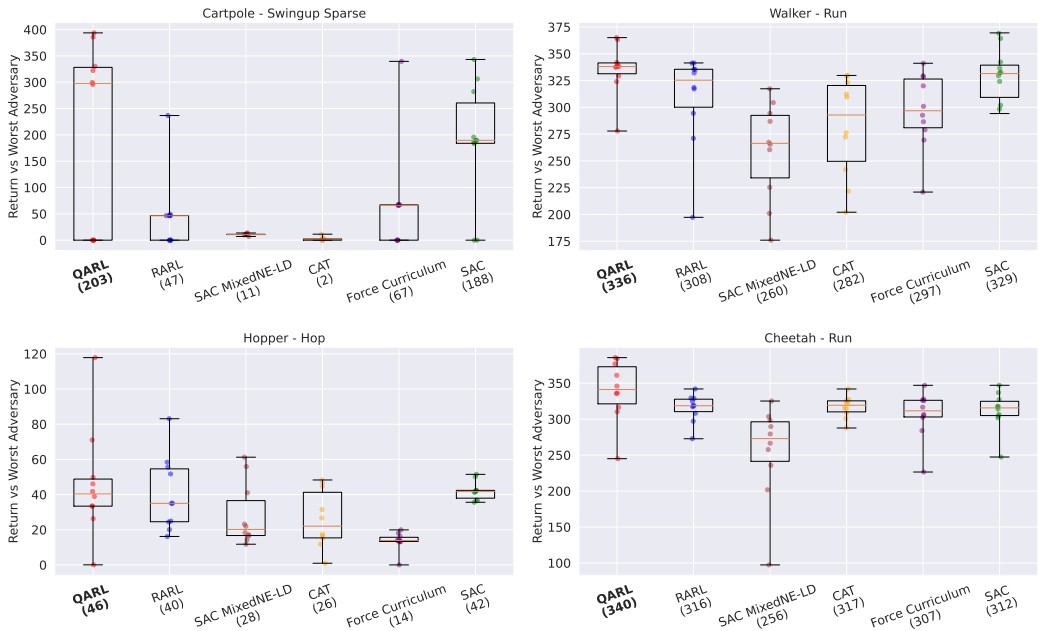

Figure 1: Performance of QARL and baselines on MuJoCo control problems (see title of boxplots), evaluated at the end of training against an adversary trained against the frozen trained protagonist. The number next to the name of each algorithm is the average performance across 10 seeds.

with $\lambda, \eta \geq 0$. While the KL-divergences can be computed exactly using the closed-form formula of the gamma distribution, the expected performance $\mathbb{E}_{p_{\boldsymbol{\omega}}(\alpha)}[J_{\mu,\nu}(\boldsymbol{\omega})]$ needs to be approximated with an importance-sampled Monte-Carlo estimate

$$\mathbb{E}_{p_{\boldsymbol{\omega}}(\alpha)}[J_{\mu,\nu}(\boldsymbol{\omega})] \approx \frac{1}{M} \sum_{i=1}^{M} \frac{p_{\boldsymbol{\omega}}(\alpha_i)}{p_{\boldsymbol{\omega}'}(\alpha_i)} \tilde{V}(s_{0,i}, \alpha_i), \tag{12}$$

where $\tilde{V}(s_{0,i}, \alpha_i)$ is an estimated value function obtained by collecting trajectories starting from $M$ initial states $s_{0,i}$, with the protagonist and the adversary following their policies $\mu$ and $\nu$ both conditioned on the temperature $\alpha_i$ of the adversary. Algorithm 1 summarizes our Quantal Adversarial RL (QARL) algorithm. Although not strictly necessary, we condition the policy of the protagonist on the temperature of the adversary (line 7 and 16) for better adaptation. We also clarify that while the temperature of the adversary follows our automatic curriculum generation scheme, the protagonist follows the temperature scheduling proposed by SAC (Haarnoja et al., 2018b) (line 20).

## 5 EXPERIMENTAL RESULTS[2]

**MuJoCo control** We consider a broad set of MuJoCo control problems (Todorov et al., 2012) from the DeepMind Control Suite (Tunyasuvunakool et al., 2020), and compare our method to RARL (Pinto et al., 2017) and recently proposed enhancements that address the complexity of the saddle point optimization problem of RARL. Namely, we consider an extension of RARL that uses Langevin dynamics (MixedNE-LD) to escape local optima (Kamalaruban et al., 2020), and Curriculum Adversarial Training (CAT) (Sheng et al., 2022) that applies a hand-crafted curriculum scheme on the strength of the adversary. Additionally, we introduce and evaluate a method, that we denote as Force-Curriculum, that applies our automatic curriculum scheme over the maximum strength of the adversary instead of its rationality (details in Appendix A). In contrast with the theoretical formulation for QARL, Force-Curriculum has the limit of having a heuristic nature and hinges on the ability to change the action boundaries of the adversary during training. Nevertheless, given its simplicity and intuitive motivation, we find Force-Curriculum an interesting additional baseline. Finally, we compare to regular SAC trained against no adversary as a reference (Haarnoja et al., 2018b).

[2]Code available at `https://github.com/AryamanReddi99/quantal-adversarial-rl`

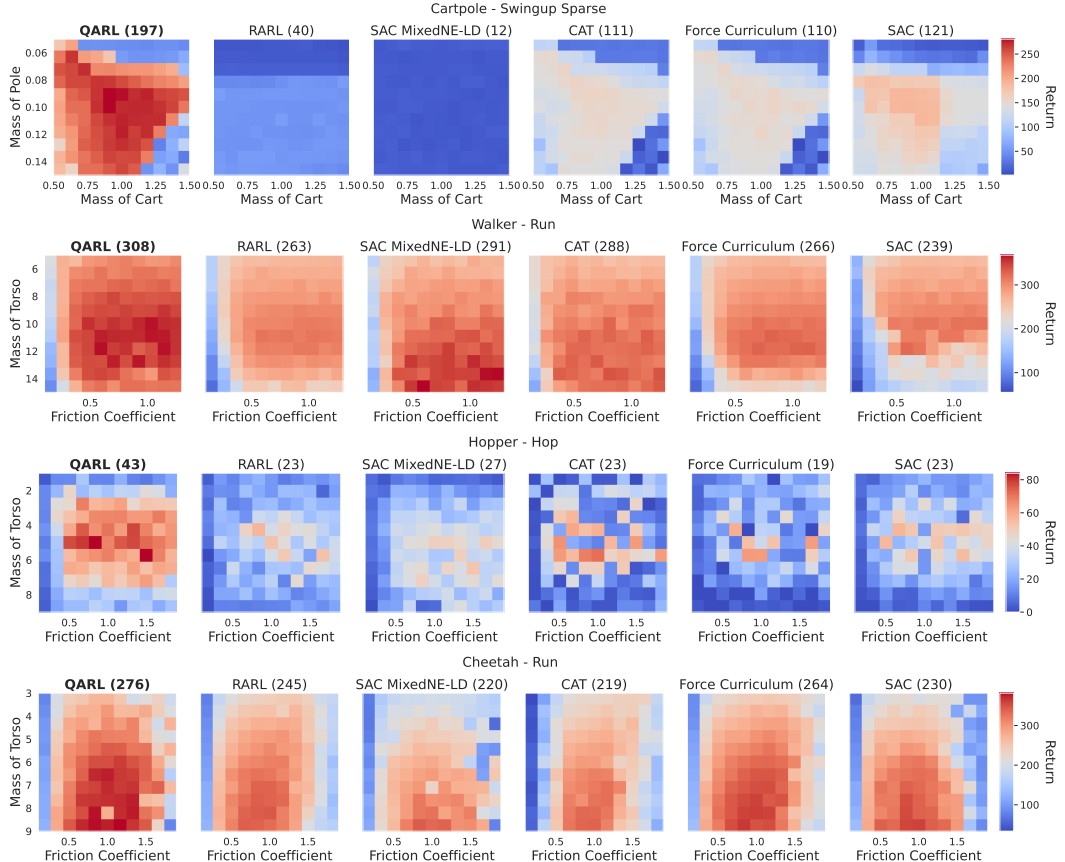

Figure 2: Robustness analysis of QARL and related baselines on MuJoCo swingup and locomotion problems (see title of heatmaps). Each heatmap shows the performance obtained for varying properties of the environment, described in the $x - y$ axes. The number next to the name of each algorithm is the average performance across 10 seeds.

In each environment, the adversary exerts forces on the protagonist to disrupt its performance. As done in Pinto et al. (2017); Kamalaruban et al. (2020), we consider environment-specific disruptions with high strength to test the reliability of considered adversarial algorithms under intense destabilization (details in Appendix B). To adequately assess the performance of the protagonist in a minimax sense, we fix the protagonist obtained at the end of training and subsequently train an adversary against it, which is then used at test time. Moreover, we evaluate the robustness of the trained protagonists to varying test conditions, such as mass

| Algorithm | Performance | Robustness |
|---|---|---|
| SAC MixedNE-LD | $-16.6 \pm 0.2\%$ | $+20.0 \pm 1.1\%$ |
| RARL | $-5.7 \pm 0.3\%$ | $+10.1 \pm 0.6\%$ |
| Force-Curriculum | $-14.7 \pm 0.3\%$ | $-3.3 \pm 0.6\%$ |
| CAT | $-18.0 \pm 0.2\%$ | $+11.7 \pm 0.8\%$ |
| QARL (ours) | $+\mathbf{4.2} \pm \mathbf{0.5}\%$ | $+\mathbf{48.7} \pm \mathbf{1.2}\%$ |

Table 1: Performance and robustness percentage improvement w.r.t. SAC for QARL and baselines in 15 MuJoCo control problems.

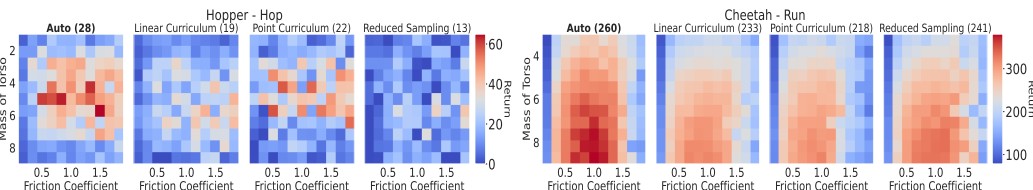

Figure 3: Robustness analysis over different curricula for QARL on two locomotion problems.

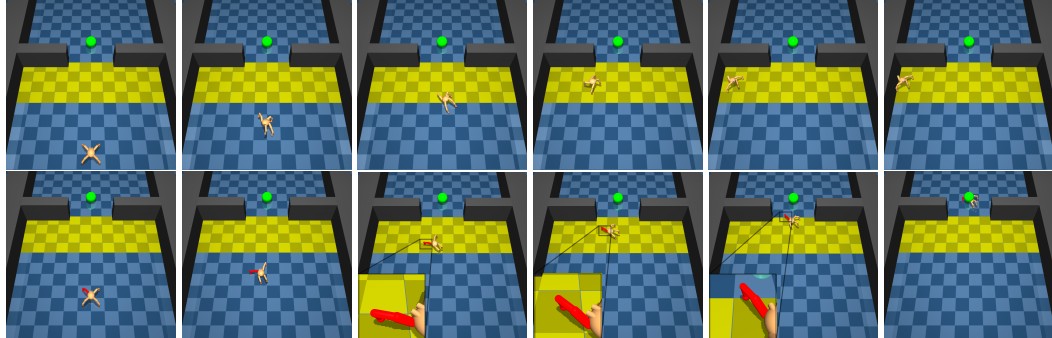

Figure 4: Trajectories of the quadruped agents trained using RARL (top) and QARL (bottom) in the maze environment. The adversary can blow wind from right to left in the area highlighted in yellow. The goal position is marked as a green sphere. The leg that the Quadruped uses to anchor when using the policy obtained with QARL is highlighted in red.

and friction, against no adversary. We conduct this evaluation on 15 MuJoCo control problems – a selection of them is shown in Figures 1 and 2, while an overall assessment is shown in Table 1 (details in Appendix C). QARL outperforms the baselines in average performance and robustness, thus evincing a superior ability to counteract the adversary. While more robust across test conditions than SAC, the other baselines suffer in nominal performance due to the highly destabilizing adversarial forces that hinder learning. The non-curriculum methods (RARL and MixedNE-LD) perform notably poorly for the environment 'Cartpole - Swingup Sparse', in which the sparse reward setting enables the adversary to easily hinder the protagonist. In Figure 3, we also study the effectiveness of our automatic curriculum generation method in terms of robustness compared to other simpler curriculum schemes. Linear Curriculum refers to a basic linear annealing of $\alpha$. Point Curriculum consists of a variant of the optimization problem (10) where $\alpha$ is a scalar value instead of being sampled from a distribution. Finally, Reduced Sampling applies our automatic curriculum generation method, sampling only a single value at each iteration.

**Maze navigation** We further analyze QARL in a navigation problem with a MuJoCo quadruped agent (Tunyasuvunakool et al., 2020) that requires learning locomotion skills to reach a goal while contrasting an adversary blowing wind. Figure 4 shows the environment and behavior of QARL against RARL, evincing that RARL collapses to a suboptimal policy while QARL is able to solve the problem by learning a gait where one leg is used to anchor to the ground while using the others to move towards the goal. Moreover, we study the robustness of the obtained policy for QARL and baselines against forces obtained by multiplying the one in training by a relative force ranging from 0 to 4. (Figure 5). We observe that QARL can resist forces 4 times stronger than the nominal one, while RARL and the other baselines are not able to, with the exception of Force-Curriculum which sometimes succeeds.

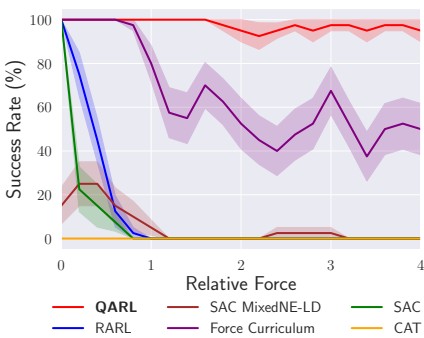

Figure 5: Robustness analysis for wind forces in the maze environment. Relative force is w.r.t. the one in training.

## 6  CONCLUSION

We introduced Quantal Adversarial Reinforcement Learning (QARL), a method leveraging the connection between maximum entropy RL and bounded rationality, to facilitate the optimization of the complex saddle point optimization problem tackled by adversarial RL. Our method is backed up by a solid theoretical foundation and extensive empirical validation on a broad set of MuJoCO control problems. By combining the benefits of curriculum and adversarial learning, QARL reduces the amount of time and energy to robustify agents, which is particularly desirable for achieving adaptive behavior in real-world applications, ranging from autonomous vehicles to assistive robotics.

## ACKNOWLEDGMENTS

This work was funded by the German Federal Ministry of Education and Research (BMBF) (Project: 01IS22078). This work was also funded by Hessian.ai through the project 'The Third Wave of Artificial Intelligence – 3AI' by the Ministry for Science and Arts of the state of Hessen. We acknowledge the grant 'Einrichtung eines Labors des Deutschen Forschungszentrum für Künstliche Intelligenz (DFKI) an der Technischen Universität Darmstadt' of the Hessisches Ministerium für Wissenschaft und Kunst.

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

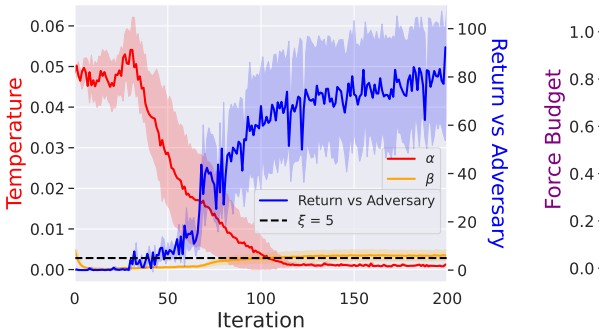 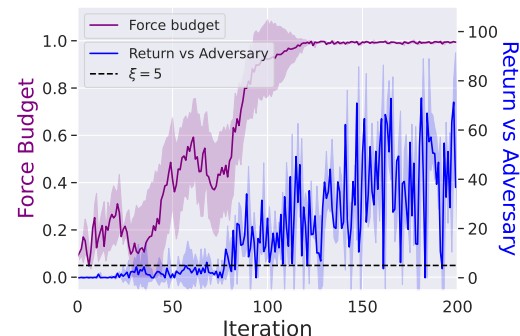

(a) QARL temperature & training performance.  (b) Force-Curriculum budget & training performance.

Figure 6: Progression of the (a) adversary temperature ($\alpha$) and protagonist temperature ($\beta$) of QARL, and (b) adversary force budget of the Force-Curriculum algorithm during training compared to average return against a trained adversary in the "Hopper - Hop" environment. The performance threshold $\xi = 5$ is also indicated. Results are shown over 10 seeds.

## A  FORCE-CURRICULA FOR ADVERSARIAL REINFORCEMENT LEARNING

In addition to the adversarial RL baselines implemented, we also find it worthwhile to compare QARL to a conceptually similar curriculum algorithm over the maximum strength of the adversary, which we refer to as "Force-Curriculum" (Algorithm 2). We follow a similar curriculum scheme to QARL, whereby force budgets are sampled from a gamma distribution which is updated according to the performance of the protagonist agent. The curriculum is enforced by clipping the magnitude of the adversary's actions in each dimension according to the sampled force budget $f$ (Line 9 and 19 in Algorithm 2). Figure 7 shows the evolution of the gamma distribution of the force budget $f$ of the adversary $p_{\boldsymbol{\omega}}(f)$.

The force budget distribution has an initial mean near $0$, preventing the adversary from creating large disturbances during early training. As training progresses, this force budget distribution shifts to higher values to enable the adversary to use its full nominal strength against the now-trained protagonist agent. The idea of gradually raising the strength of the adversary is also implemented in Curriculum Adversarial Training (CAT) (Sheng et al., 2022), which, however, uses a linear curriculum as opposed to our automatic curriculum generation scheme. Figure 6 compares the training progression of QARL to that of Force-Curriculum in the "Hopper - Hop" environment.

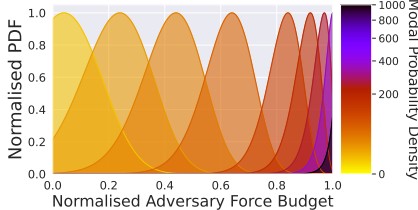

Figure 7: Evolution of adversary force budget $f$ distribution.

Figure 6a shows the progression of the adversary temperature in QARL overlaid with the performance of the protagonist agent against the adversary. Similarly, Figure 6b shows the progression of the adversary force budget in the Force-Curriculum algorithm overlaid with the performance of the protagonist agent against the adversary. Both plots indicate the performance threshold $\xi$ above which the curriculum would progress to a more difficult adversary. We set $\xi = 5$ for the "Hopper - Hop" environment, as performance below this value indicates a catastrophically poor protagonist policy. It is observed in Figure 6a that the temperature of the adversary in QARL is decreased when the performance of the protagonist is above $\xi$, and increased or maintained when the performance of the protagonist is below $\xi$ (note that the precise crossing points of the performance with the performance threshold cannot be matched $1 : 1$ with the curriculum temperature behavior due to some variance in the plots). Likewise, it can be seen in Figure 6b that the adversary force budget $f$ in the Force-Curriculum algorithm is increased when the performance of the protagonist is above $\xi$ and decreased or maintained when the performance of the protagonist is below $\xi$. We remark that "Hopper - Hop" is chosen for this comparison as it is prone to instability, being a perfect candidate to evince the benefit of QARL.

---

**Algorithm 2** Force-based curriculum adversarial reinforcement learning (Force-Curriculum)

---

1: **Input:** Initial critic parameters $\psi_1^1, \psi_2^1, \bar{\psi}_1^1, \bar{\psi}_2^1, \psi_1^2, \psi_2^2, \bar{\psi}_1^2, \bar{\psi}_2^2$; initial actor parameters $\phi^1, \phi^2$;
   $\mathcal{D}^1 = \emptyset, \mathcal{D}^2 = \emptyset$; initial force distribution parameters $\boldsymbol{\omega}$; performance lower bound $\xi$; #sampled
   forces $N$; #Monte-Carlo rollouts $M$; initial temperatures $\alpha, \beta$; initial force budget $f$; #steps $T$.
2: **for** each iteration **do**
3:     $\bar{\psi}_1^1 \leftarrow \psi_1^1, \bar{\psi}_2^1 \leftarrow \psi_2^1, \bar{\psi}_1^2 \leftarrow \psi_1^2, \bar{\psi}_2^2 \leftarrow \psi_2^2$                ▷ Target networks update
4:     $f_{1,\ldots N} \sim p_{\boldsymbol{\omega}}(f))$             ▷ Sampling of force budgets of the adversary
5:     **for** $i = 1, \ldots, N$ **do**
6:        **for** $t = 1, \ldots, T$ **do**           ▷ Transitions collection for the adversary
7:           $a_t^1 \sim \mu_{\phi^1}(a_t^1 | s_t, f_i)$
8:           $a_t^2 \sim \nu_{\phi^2}(a_t^2 | s_t, f_i)$
9:           $a_t^2 = CLIP(a_t^2, -f_i, f_i)$
10:          $s_{t+1} \sim \mathcal{P}(s_{t+1} | s_t, a_t^1, a_t^2)$
11:          $\mathcal{D}^2 = \mathcal{D}^2 \cup \{\langle s_t, a_t^2, r_t, s_{t+1}, f_i \rangle\}$
12:          $\psi_1^2, \psi_2^2, \phi^2, \alpha \leftarrow \text{ADVERSARY\_UPDATE}(\psi_1^2, \psi_2^2, \phi^2, \alpha, \mathcal{D}^2)$
13:        **end for**
14:     **end for**
15:     **for** $i = 1, \ldots, N$ **do**
16:        **for** $t = 1, \ldots, T$ **do**           ▷ Transitions collection for the protagonist
17:           $a_t^1 \sim \mu_{\phi^1}(a_t^1 | s_t, \alpha_i)$
18:           $a_t^2 \sim \nu_{\phi^2}(a_t^2 | s_t, \alpha_i)$
19:           $a_t^2 = CLIP(a_t^2, -f_i, f_i)$
20:          $s_{t+1} \sim \mathcal{P}(s_{t+1} | s_t, a_t^1, a_t^2)$
21:          $\mathcal{D}^1 = \mathcal{D}^1 \cup \{\langle s_t, a_t^1, r_t, s_{t+1}, f_i \rangle\}$
22:          $\psi_1^1, \psi_2^1, \phi^1, \beta \leftarrow \text{PROTAGONIST\_UPDATE}(\psi_1^1, \psi_2^1, \phi^1, \beta, \mathcal{D}^1)$
23:        **end for**
24:     **end for**
25:     Update $\boldsymbol{\omega}$ optimizing (10), estimating $\mathbb{E}_{p_{\boldsymbol{\omega}}(f)}[J_{\mu,\nu}(\boldsymbol{\omega})]$ with $M$ rollouts
26: **end for**
27: **Return:** Trained actor parameters $\phi^1, \phi^2$

---

# B    EXPERIMENTAL DETAILS

## B.1    AGENTS

The majority of agents deployed in the experiments (except the adversary policy in CAT) are SAC agents and variations thereupon, allowing us to leverage the temperature parameter of SAC for the bounded rationality curriculum of QARL; SAC has been then chosen similarly for the other algorithms to enable a fair comparison. The hyperparameters of the adversarial agents across all environments are shown in Table 2. Note that the temperature-related hyperparameters (initial temperature, temperature learning rate, target entropy) do not apply to the adversary used in QARL as they are dictated by the automatically generated curriculum. For the SAC MixedNE-LD agent, additional hyperparameters are chosen based on tuning and prior knowledge from Kamalaruban et al. (2020). The adversary used in CAT (Sheng et al., 2022) has separate hyperparameters as it does not use a neural network implementation, but rather performs a simple gradient descent in the action space.

## B.2    ENVIRONMENTS

The MuJoCo environments (Todorov et al., 2012) discussed in Section 5 are standard implementations from the DeepMind Control Suite (Tunyasuvunakool et al., 2020) modified for an adversarial setting. The standard environment states are used as inputs to both agents. In each environment, adversarial actions and force magnitudes are chosen to elicit robust agent behavior. The adversarial action spaces are specifically chosen to be different from those of the protagonist agent in order to exploit domain knowledge, as done in RARL (Pinto et al., 2017). Generally, a smaller adversary maximum force is used in environments that are highly sensitive to adversarial disturbances such as "Acrobot", as higher forces cause a total collapse of the protagonist policy. For each environment,

Table 2: Agent hyperparameters

| Hyperparameter | Value |
|---|---|
| *Shared (SAC and SAC MixedNE-LD)* | |
| # hidden layers (all networks) | 3 |
| # hidden units per layer (all networks) | 256 |
| nonlinearity | ReLU |
| critic optimiser | Adam |
| critic learning rate | $3 \cdot 10^{-4}$ |
| actor learning rate | $1 \cdot 10^{-4}$ |
| initial replay memory size | $3 \cdot 10^3$ |
| max replay memory size | $1 \cdot 10^6$ |
| warmup transitions | $5 \cdot 10^3$ |
| batch size | 256 |
| target smoothing coefficient ($\tau$) | $5 \cdot 10^{-3}$ |
| target update interval | 1 |
| policy log std bounds | $[-20, 2]$ |
| initial temperature | $5 \cdot 10^{-3}$ |
| temperature learning rate | $3 \cdot 10^{-4}$ |
| target entropy | -dim($\mathcal{A}$) |
| *SAC* | |
| actor optimiser | Adam |
| *SAC MixedNE-LD* | |
| adversary influence $\delta$ | 0.1 |
| actor optimiser | SGLD |
| thermal noise ($\sigma_t$) | $10^{-3} \times (1 - 5 \times 10^{-5})^t$ |
| RMSProp parameter $\alpha$ | 0.999 |
| RMSProp parameter $\epsilon$ | $10^{-8}$ |
| *CAT MAS Adversary* | |
| gradient descent learning rate | 3 |
| gradient descent step limit | 25 |
| gradient descent convergence threshold $\epsilon$ | $10^{-3}$ |
| disturbance $L^p$-norm | 2 |

Table 3: Environment-specific parameters

| Environment | Adversary max force | $\xi$ | Adversary action space (size) |
|---|---|---|---|
| Acrobot | 0.0005 | 1 | 2D forces on each arm (4) |
| Ball in Cup | 0.1 | 10 | 2D force on ball (2) |
| Cartpole | 0.005 | 10 | 2D force on pole (2) |
| Cheetah | 1.0 | 40 | 2D force on feet & torso (6) |
| Hopper | 1.0 | 5 | 2D force on foot & torso (4) |
| Pendulum | 0.005 | 10 | 2D force on pole (2) |
| Quadruped - Run | 10 | 50 | 3D force on torso (1) |
| Quadruped - Maze | 600 | 2000 | 1D force on all bodies (1) |
| Reacher | 0.1 | 10 | 2D force on arm (2) |
| Walker | 1.0 | 10 | 2D forces on feet (4) |

the adversary forces are chosen carefully so as to be large enough to beget agent robustness and generalisation while posing a challenge to the protagonist. These environment-specific parameters are shown in Table 3, while the discount factor and horizon are set respectively to $0.99$ and $500$ for all environments. Apart from this, the MuJoCo environments are left unmodified, with two exceptions: (a) the pole of the "Pendulum" environment is given a default mass of $0.1$ instead of $0$ in order to test the robustness of the protagonist to changing pole mass, and (b) the quadruped agent is trained in a maze environment where the wind force of the adversary is distributed evenly across all of its body when its torso enters the effective zone of the adversary (the yellow area in Figure 4). Note that $\xi$ (performance threshold) values shown in Table 3 apply to both QARL and Force-Curriculum. Note also that the adversary action spaces in Table 3 do not apply to CAT (Sheng et al., 2022) and MixedNE-LD (Kamalaruban et al., 2020), as these algorithms use a shared action space for the protagonist and adversary.

### B.2.1 QUADRUPED

The quadruped maze environment discussed in Section 5 is a modified version of the default quadruped provided in the DeepMind Control Suite (Tunyasuvunakool et al., 2020). The state $\mathbf{s} \in \mathbb{R}^{80}$ contains the default quadruped observations (egocentric state, torso velocity, torso upright state, IMU measurements, and body force torques) as well as the 2D displacement vector in the $X - Z$ plane between the torso of the quadruped and the goal. The reward function for the quadruped is:

$$r(\mathbf{s}, \mathbf{a}) = \langle \mathbf{z}_{torso} , \mathbf{z}_{global} \rangle \times 5 \times e^{-0.2||\mathbf{xz}_{goal} - \mathbf{xz}_{torso}||}. \tag{13}$$

The term $\langle \mathbf{z}_{torso}, \mathbf{z}_{global} \rangle$ is the inner product of the quadruped's torso's z-axis and the global z-axis, acting as a measure of how upright the quadruped is, which improved locomotion stability in practice. The exponential term $||\mathbf{xz}_{goal} - \mathbf{xz}_{torso}||$ is the Euclidian distance between the goal and the torso of the quadruped in the $X - Z$ plane, encouraging the quadruped to move towards the goal. Finally, the scale factors $5$ and $0.2$ are used to shape the exponential reward characteristic in order to sufficiently encourage the quadruped to move forward from its starting position. If the quadruped is flipped over during training (indicated by $\langle \mathbf{z}_{torso}, \mathbf{z}_{global} \rangle < 0$), the episode is terminated in order to avoid collecting junk transitions as it is hard for the quadruped to recover from this state.

### B.3 ALGORITHMS

The details of the algorithms investigated are shown in Table 4. Note that the gamma distribution parameters shown apply to both QARL and Force-Curriculum. The algorithm parameters for Curriculum Adversarial Training (CAT) in Table 4 denote the iterations between which the gradient of the linear force budget curriculum would be constant and non-zero, i.e., over the course of 200 total iterations, the force budget in CAT would simply be 0 before 40 iterations, full force after 160 iterations, and linearly interpolated for all iterations in between. These values are found to be heuristically effective and are based in part on previous experiments using manual curricula.

Table 4: Algorithm hyperparameters

| Hyperparameter | Value |
|---|---|
| *Shared* | |
| # iterations | 200 |
| # episodes per agent per iteration | 5 |
| # evaluation rollouts per iteration | 10 |
| *QARL & Force-Curriculum* | |
| initial gamma distribution concentration $k_{initial}$ | 50 |
| target gamma distribution concentration $k_{target}$ | 1 |
| fixed gamma distribution rate $\theta$ | 1000 |
| $D_{KL}$ constraint ($\epsilon$) | 0.5 |
| # rollouts needed for update | 30 |
| *CAT* | |
| curriculum start iteration | $0.2 \times$ # iterations |
| curriculum end iteration | $0.8 \times$ # iterations |

## B.4 HARDWARE AND SOFTWARE

The experiments are carried out on a computational cluster with 64GB of RAM and an AMD Ryzen 9 16-Core processor. The algorithms investigated are implemented using the Mushroom RL (D'Eramo et al., 2021) library, which is also used for the implementation of all agents and adversarial environment wrappers.

## C ADDITIONAL RESULTS

### C.1 QARL AND BASELINES PERFORMANCE AND ROBUSTNESS

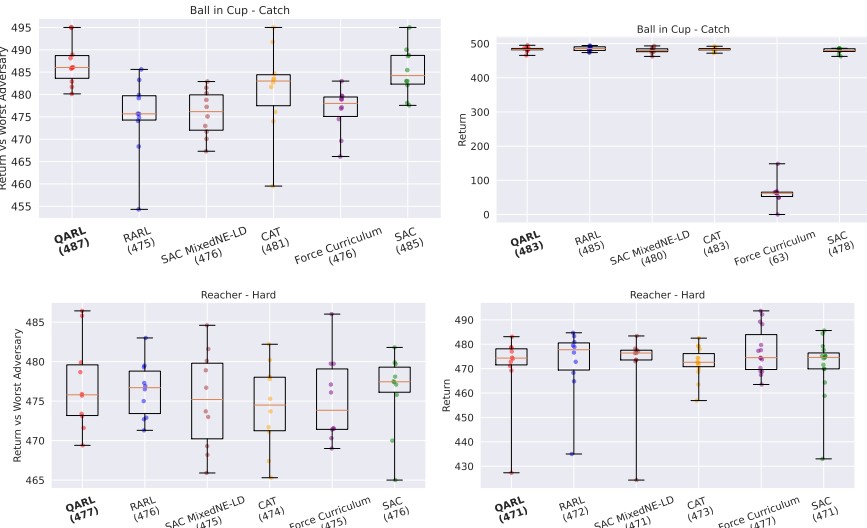

Figure 8: Performance of QARL and baselines on Ball-in-cup and Reacher (see the title of each boxplot), evaluated at the end of training against an adversary trained against the frozen trained protagonist (left column) and without an adversary (right column). The number next to the name of each algorithm is the average performance across 10 seeds.

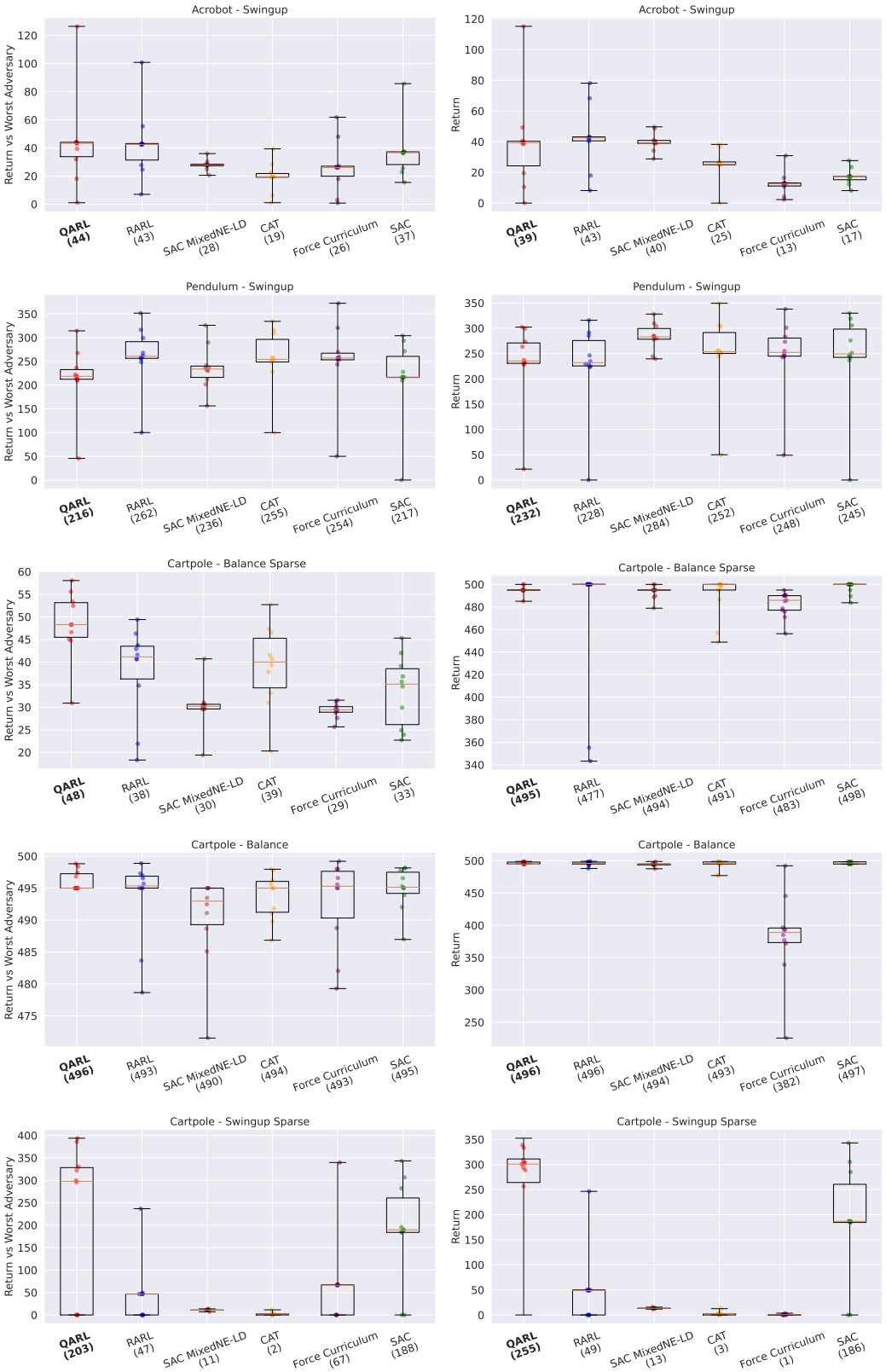

Figure 9: Performance of QARL and baselines on balancing and swing-up control problems (see the title of each boxplot), evaluated at the end of training against an adversary trained against the frozen trained protagonist (left column) and without an adversary (right column). The number next to the name of each algorithm is the average performance across 10 seeds.

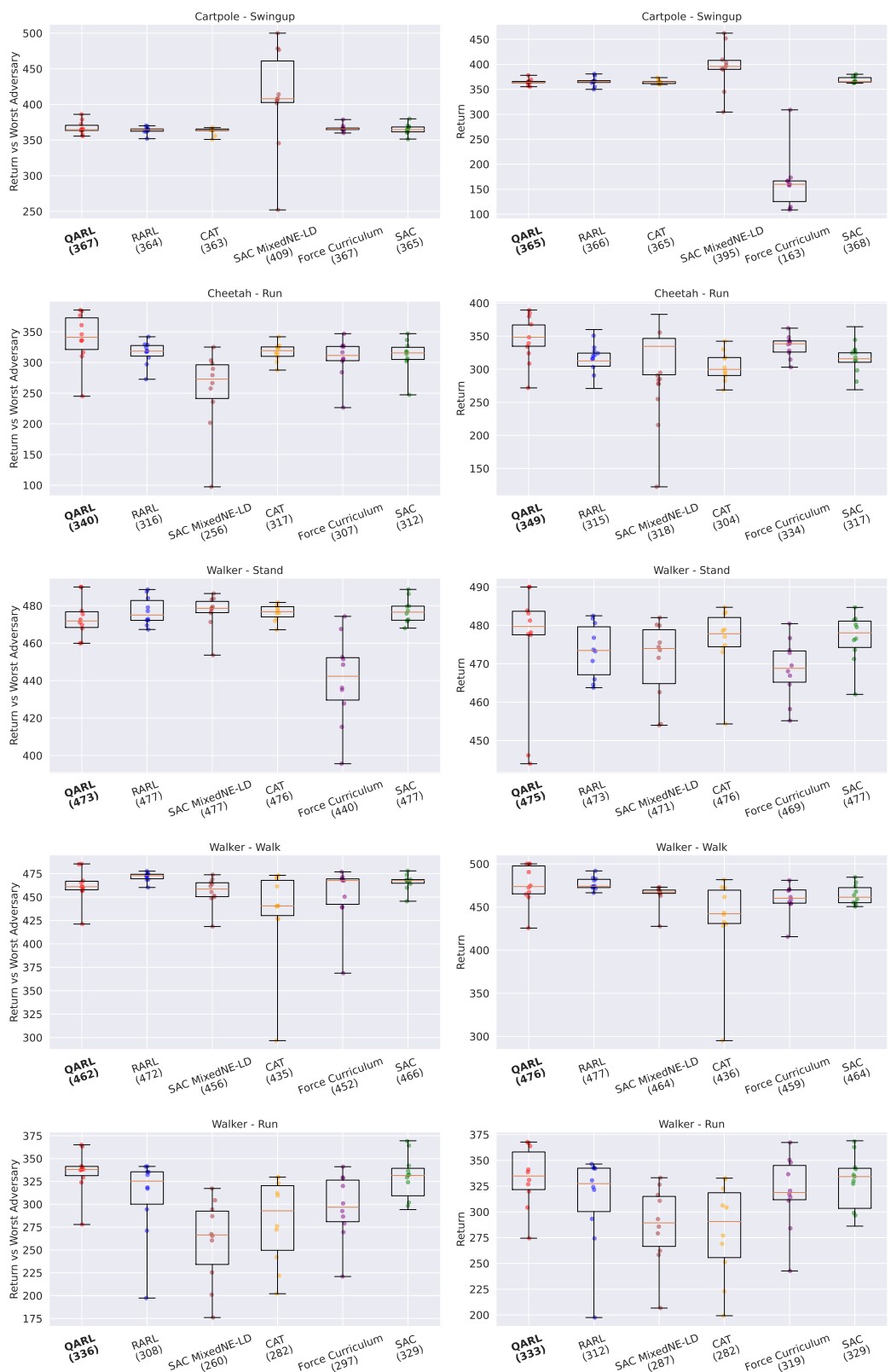

Figure 10: Performance of QARL and baselines on swing-up and locomotion problems, i.e., Cheetah and Walker (see the title of each boxplot), evaluated at the end of training against an adversary trained against the frozen trained protagonist (left column) and without an adversary (right column). The number next to the name of each algorithm is the average performance across 10 seeds.

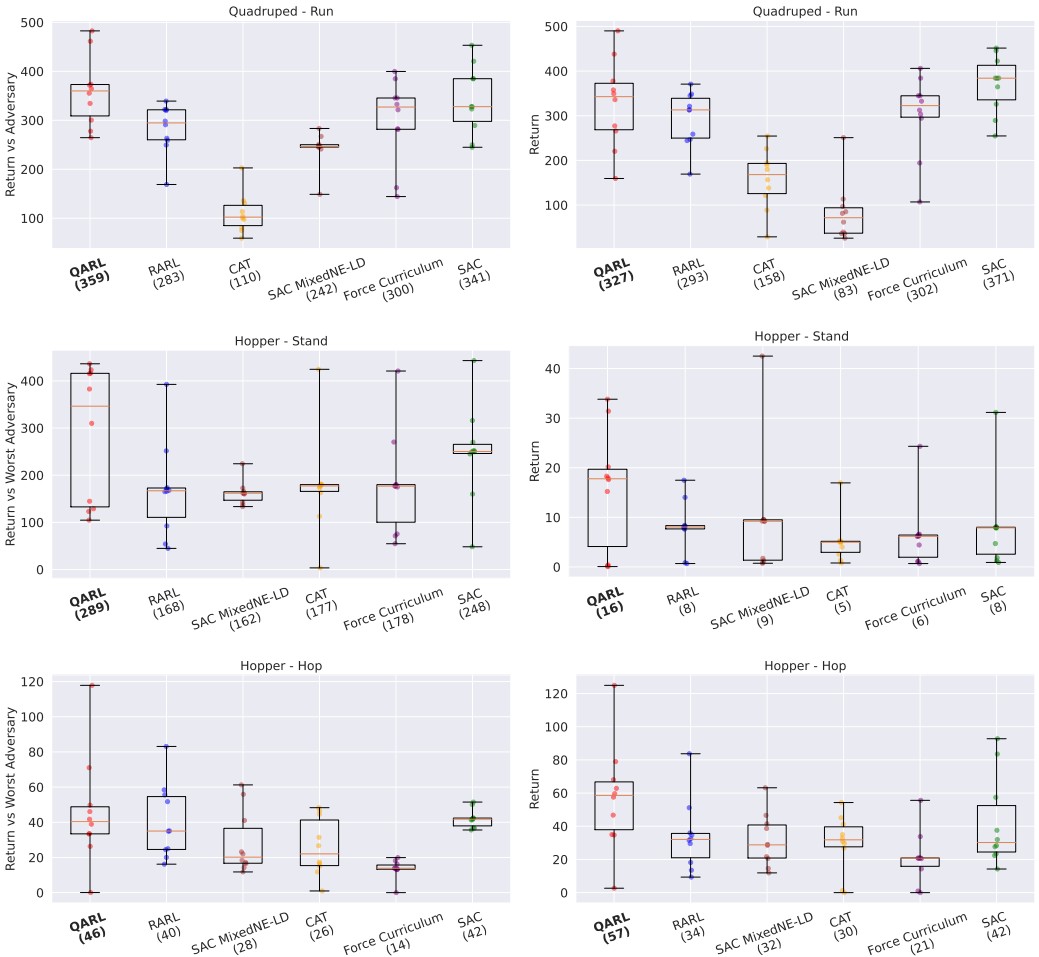

Figure 11: Performance of QARL on locomotion problems, i.e., Quadruped and Hopper (see the title of each boxplot), evaluated at the end of training against an adversary trained against the frozen trained protagonist (left column) and without an adversary (right column). The number next to the name of each algorithm is the average performance across 10 seeds.

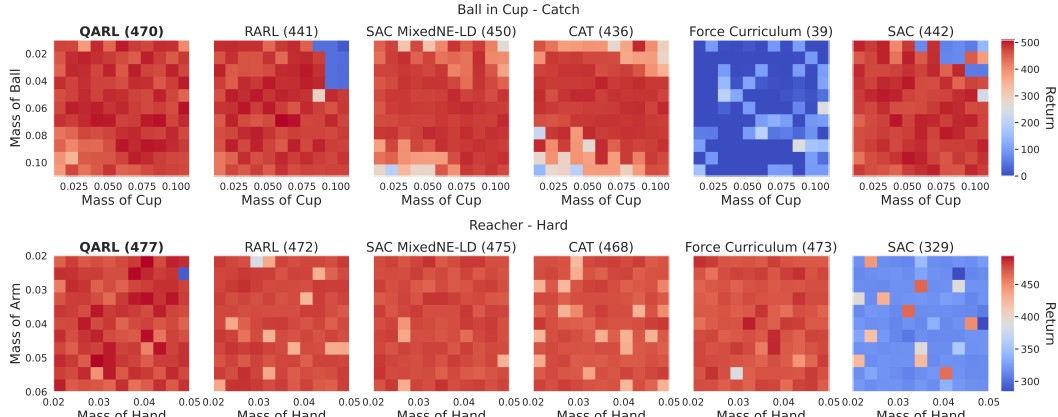

Figure 12: Robustness analysis on Ball-in-cup and Reacher (see the title of each heatmap set). Each heatmap shows the performance obtained for varying properties of the environment, described in the $x - y$ axes. The number next to the name of each algorithm is the average performance across 10 seeds.

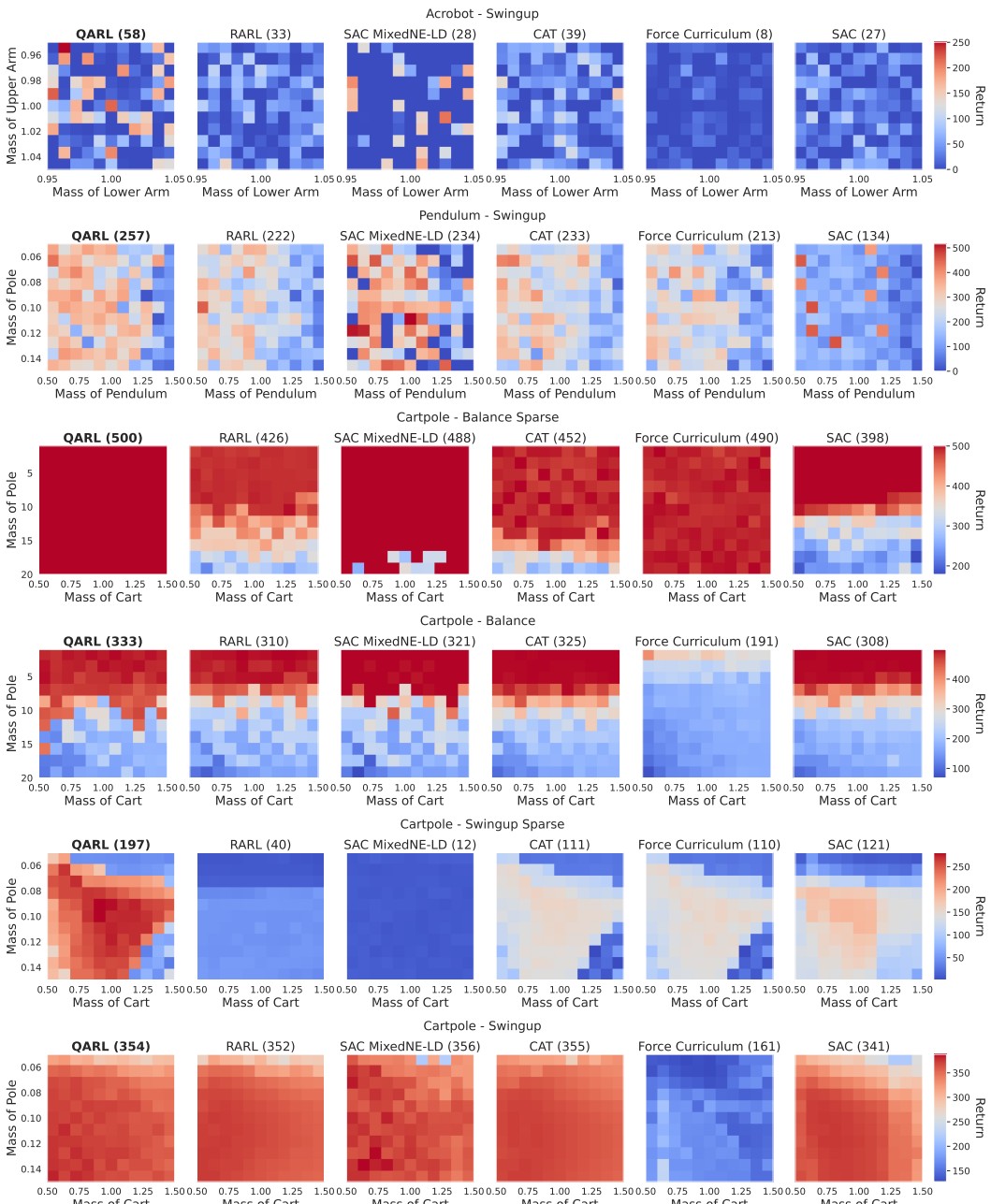

Figure 13: Robustness analysis of QARL and baselines on balancing and locomotion problems, i.e., Cartpole, Cheetah, Walker, and Hopper (see the title of each heatmap set). Each heatmap shows the performance obtained for varying properties of the environment, described in the $x - y$ axes. The number next to the name of each algorithm is the average performance across 10 seeds.

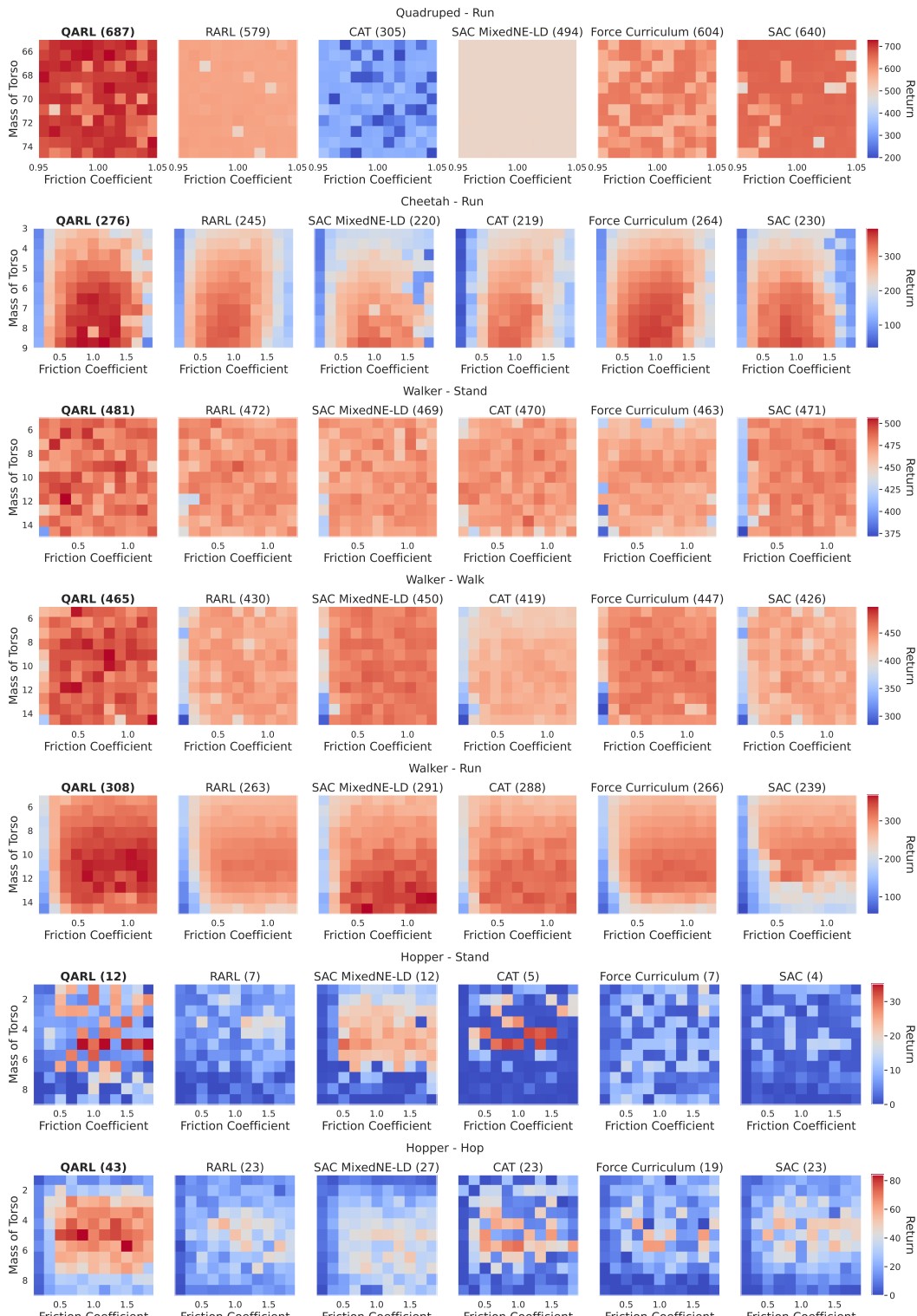

Figure 14: Robustness analysis of QARL and baselines on locomotion problems, i.e., Quadruped, Cheetah, Walker, and Hopper (see the title of each heatmap set). Each heatmap shows the performance obtained for varying properties of the environment, described in the $x - y$ axes. The number next to the name of each algorithm is the average performance across 10 seeds.

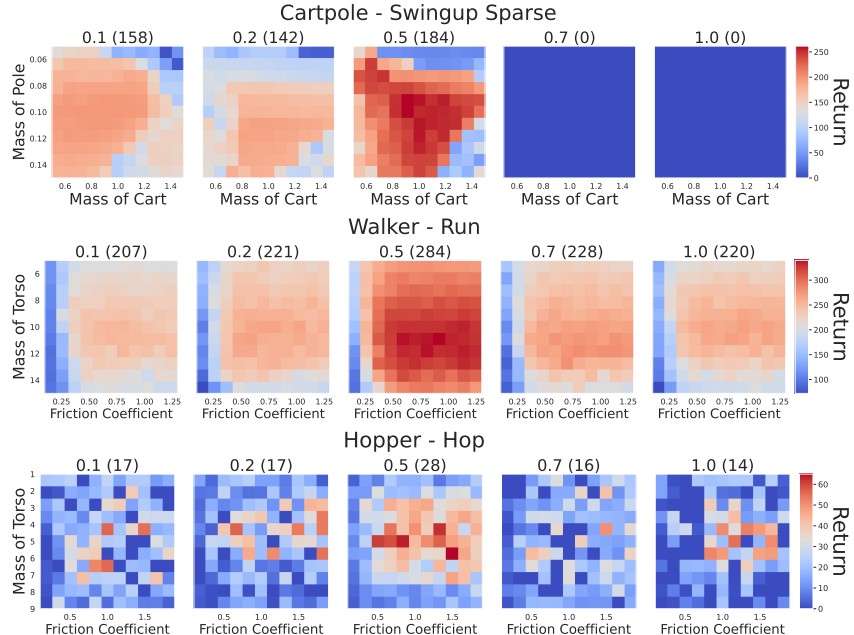

Figure 15: Robustness analysis of QARL trained with several values of $D_{KL}$ constraint ($\epsilon$) on three selected MuJoCo control problems. Each heatmap shows the performance obtained for varying properties of the environment, described in the $x-y$ axes. Each heatmap is titled with the value of $\epsilon$ used during training followed by the average performance across 10 seeds in brackets.

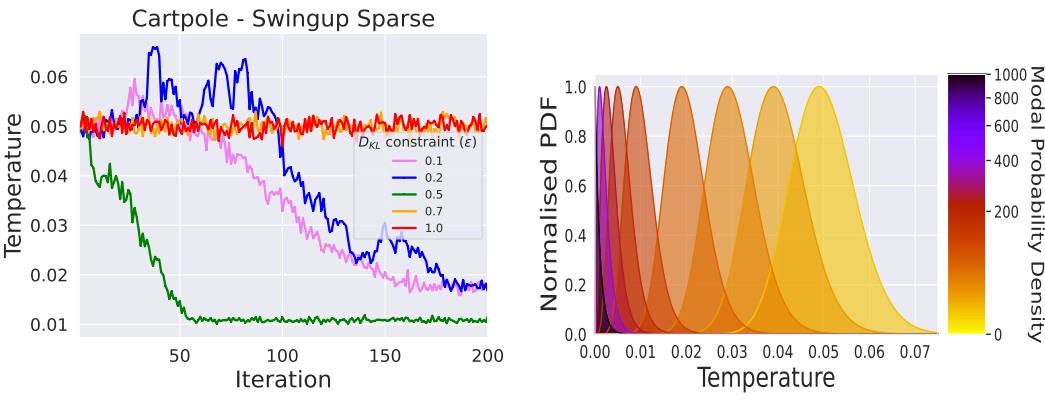

(a) Mean temperature across different $D_{KL}$ constraint.   (b) Evolution of temperature distribution.

Figure 16: (a) Mean temperature of QARL adversary during training in the 'Cartpole - Swingup' environment across several values of $D_{KL}$ constraint ($\epsilon$). (b) Evolution of adversary temperature distribution from high (light) to low (dark) as the curriculum progresses.

## C.2   QARL PROGRESSION AND ABLATION

Figure 15 compares the robustness of agents trained with QARL across several values of the $D_{KL}$ constraint ($\epsilon$) in Equation (10). It is seen that too-high and too-low values of $\epsilon$ result in suboptimal robustness, indicating that an intermediate value is generally preferable. Figure 16a also suggests that an intermediate value for $\epsilon$ allows the temperature curriculum to progress at an appropriate pace. Lower values of $\epsilon$ cause the automatic curriculum to progress too slowly to obtain a rational adversary at the end of training. Meanwhile, higher values of $\epsilon$ result in large jumps of the temperature distribution between iterations, causing changes in adversary behavior that are too drastic for the protagonist to adapt to, curtailing agent performance, and preventing the curriculum from progress-

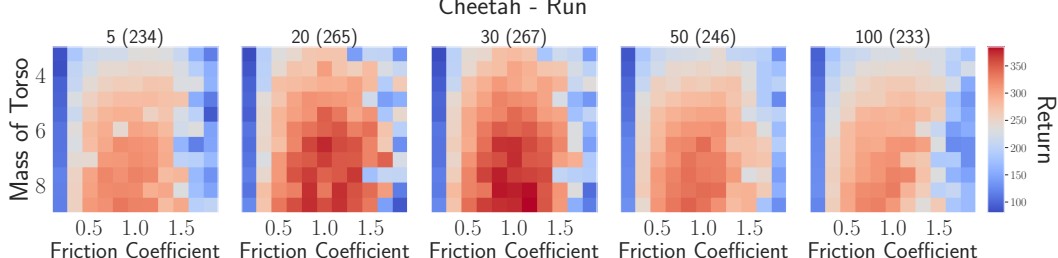

Figure 17: Robustness analysis of QARL trained with several values of performance threshold $\xi$ in the Cheetah MuJoCo environment. Each heatmap shows the performance obtained after training for varying properties of the environment, described in the x - y axes. Each heatmap is titled with the value of $\xi$ used during training followed by the average performance across 10 seeds in brackets.

ing. In practice, some initial tuning of the $D_{KL}$ constraint hyperparameter is sufficient to obtain a single value for $\epsilon$ that enables robust behavior across all environments.

Figure 16b shows the evolution of the gamma distribution of the temperature $\alpha$ of the adversary $p_\omega(\alpha)$. We start from a gamma distribution with a high mean to sample high temperatures. The initial low rationality of the adversary enables high protagonist agent performance due to the relative simplicity of the maximum entropy minimax optimisation. Progressively, the mean of the distribution approaches 0, and the variance decreases. The resulting protagonist agent is endowed with high robustness and performance despite the relative complexity of the saddle point Nash equilibrium optimisation.

Figure 17 ablates the value of the performance threshold $\xi$ in the MuJoCo Cheetah environment. It is generally found that low (but not extremely low) values of $\xi$ result in stable convergence of the curriculum, e.g. $\xi = 20$ and $\xi = 30$ in Figure 17, as these values allow the curriculum to progress as long as the protagonist is not catastrophically failing at the task. Too low values (e.g. $\xi = 5$) result in the curriculum dropping too fast, whereas too high values (e.g. $\xi = 50$, $\xi = 100$) result in the curriculum failing to progress since the agent is required to reach an unreasonable threshold.

Figure 18 tests the importance of distinct temperatures for the protagonist and adversary agents. As the aim of QARL is to robustify the protagonist agent, it attempts to solve a heterogeneous QRE as discussed in Chapter 4 of Goeree et al. (2016). We choose not to excessively bound the rationality of the protagonist so that it can reasonably solve the task at each iteration and efficiently progress to the end of the curriculum, where the robustness benefits are maximised.

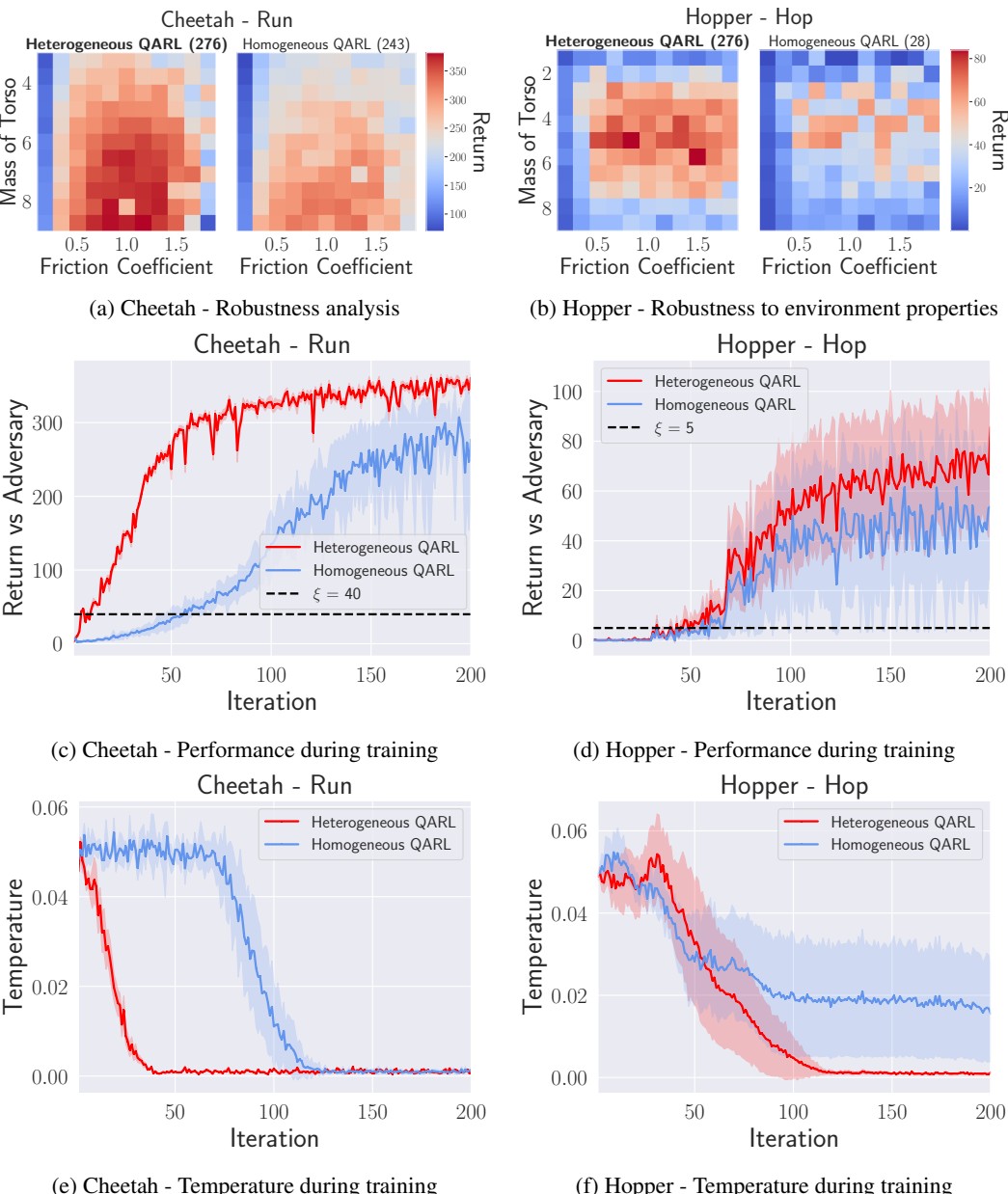

Figure 18: Ablation comparing heterogeneous QARL (ours) to homogeneous QARL over the Cheetah and Hopper environments across 10 seeds. (a) and (b) show heatmaps of performance for varying properties of the environment described in the $x - y$ axes. Each heatmap is titled with the algorithm and the average performance in brackets. (c) and (d) show the performance during training against an entropy-regularised adversary. The dashed lines indicate the performance threshold $\xi$ for each environment. (e) and (f) show the curriculum temperature during training. Note that in the homogeneous case, both agents follow the temperature as dictated by the curriculum, whereas in the heterogeneous case, the protagonist agent follows the default temperature schedule of SAC (not shown here), like that shown in Figure 6a.

