# OpenReview forum: "Robust Adversarial Reinforcement Learning via Bounded Rationality Curricula"
_ICLR.cc/2024/Conference — ICLR 2024 spotlight_

### Official Review · Reviewer_cFcL · 2023-10-28

**Soundness:** 3 good
**Presentation:** 2 fair
**Contribution:** 3 good
**Rating:** 6
**Confidence:** 4

**Summary:**

This paper proposes a robust RL framework on top of RARL, considering the entropy-regularized problem corresponds to a Quantal Response Equilibrium (QRE). With this extension, solving zero-sum games will not always face complex saddle point optimization problems. This paper is with theoretical support and numerous impressive experiments.

**Strengths:**

1. The algorithm is novel in the robust learning community.
2. The empirical results demonstrate that the proposed method outperforms the existing baselines.
3. Proposed methods are well motivated with good explainability.

**Weaknesses:**

1. Overall, how the proposed method is adapted to the real world with different action spaces between protagonist and adversary is still unclear. The detailed questions and concerns please refer to the questions section.

**Questions:**

1. Your framework is built on the top of RARL, which means that the adversarial action spaces are specifically chosen to be different from those of the protagonist agent in order to exploit domain knowledge. Could you explain more about how you set up the adversarial action spaces with the additional environments/tasks compared with the ones (e.g., halfcheetah, swimmer,  hopper ) RARL provides?

2. Is your proposed method also compatible with Noisy Robust Markov Decision Process (NR-MDP), which MixedNE-LD builds on top of? If so, what the role of $\alpha$ will be in NR-MDP? Is it just similar to the concept of $\delta$ mentioned in MixedNE-LD paper for defining the limit of the adversary?

3. How can we elaborate the concept of rationality in the experiment? Does the most rational adversary mean the strongest adversary (severity strength)? In that case, I am not sure if irrationality at the beginning represents the less strength of attack.

4. On page 2, a statement: "Conversely, in a setting where the protagonist is completely rational and the adversary is completely irrational, i.e.,
it plays only random actions...". Do you mean that the protagonist is playing only random actions? Then what action of the adversary take?

5. On page 5, a statement: "We propose to initially solve an adversarial problem with a completely irrational adversary, i.e., $\alpha \rightarrow \infty$, which results in a simpler plain maximization of the performance of the protagonist, neglecting robustness". Does it mean that we do not have any attack now?

6. Could you provide the hyperparameter tuning or how you select $\xi$ and $\epsilon$? I think they will also influence the adversarial learning process.

7. Could you please point out which script.py under mujoco_experiments is your proposed method?

---

> ### Author Response · Authors · 2023-11-17
>
> > Could you explain more about how you set up the adversarial action spaces...
>
> We thank the Reviewer for this interesting question. In the spirit of the original RARL paper [1], we aimed to define action spaces which allow even a weak adversary to sufficiently disrupt the operation of the protagonist. For instance, in swingup tasks (Pendulum, Acrobot, Cartpole) we allowed the adversary to push directly on the pole(s). We found that this led to a greater robustness in trained agents, and also highlights the need for regularization as RARL is sometimes not able to solve these sensitive environments. For the specifics of the adversarial action spaces for each environment, we point to Table 3 in Appendix B.2.
>
> > Compatible with Noisy Robust Markov Decision Process (NR-MDP), which MixedNE-LD builds on top of? If so, what the role of will be in NR-MDP? Is it just similar to the concept of mentioned in MixedNE-LD paper for defining the limit of the adversary?
>
> Thanks for raising the interesting NR-MDP framework. QARL should be compatible with the NR-MDP framework by running our curriculum method over the rationality of the adversarial mixing agent in NR-MDP while keeping the action-mixing ratio fixed. We posit that this would be more successful than running the curriculum over the action-mixing ratio directly ($\delta$ in MixedNE-LD) as our experiments suggest that our curriculum over rationality generally leads to better performance and robustness than methods which run a curriculum over adversary disturbance magnitude such as [2] and Force-Curriculum (described in Appendix A of our work).
>
> > How can we elaborate the concept of rationality in the experiment? Does the most rational adversary mean the strongest adversary (severity strength)? In that case, I am not sure if irrationality at the beginning represents the less strength of attack.
>
> > On page 5, a statement: "We propose to initially solve an adversarial problem with a completely irrational adversary, i.e., , which results in a simpler plain maximization of the performance of the protagonist, neglecting robustness". Does it mean that we do not have any attack now?
>
> We apologise for any confusion; when we refer to rationality, we refer specifically to the likelihood of playing random actions, as done in [3], not the magnitudes of the disturbances exerted by the adversary agent. A low-rationality adversary at the beginning of training will mostly play random actions, resulting in an easier objective maximization for the protagonist. In the revised version, we have corrected all mentions of 'irrational' behavior in our work with 'minimal rationality' or 'random'.
>
> > On page 2, a statement: "Conversely, in a setting where the protagonist is completely rational and the adversary is completely irrational, i.e., it plays only random actions...". Do you mean that the protagonist is playing only random actions? Then what action of the adversary take?
>
> We apologise for the unclear wording here. The pronoun 'it' in 'it plays only random actions...' refers to the adversary. We have updated the paper to make this more clear.
>
> > Could you provide the hyperparameter tuning or how you select $\xi$ and $\epsilon$?
>
> The selection of $\xi$ can be done considering the environment at hand, and the exact values used are listed in Table 3 in Appendix B.2. We find that choosing a relatively low value (as a fraction of the total achievable score for each environment) is the most reliable method for tuning $\xi$, as this progresses the curriculum under the condition that the protagonist has not catastrophically failed at solving the task at each iteration. We thank the reviewer for raising this question. We have included a new ablation study on $\xi$ in Figure 17 of Appendix C.2 which further demonstrates our reasoning.
>
> The hyperparameter $\epsilon$ is tuned along with the parameters of the initial and target temperature distributions. Initial tuning of these hyperparameters resulted in a stable set of values that worked consistently well across all the environments tested. This is thanks in part to the relationship between temperature and agent rationality remaining fairly consistent across different adversarial action spaces. The value for $\epsilon$ and other curriculum hyperparameters are listed in Table 4 of Appendix B.3.
>
> > Could you please point out which script.py under mujoco_experiments is your proposed method?
>
> The script that describes our method is:
>
> `code/src/experiments/mujoco_experiments/qarl_experiment.py`
>
> [1] Pinto, Lerrel, James Davidson, Rahul Sukthankar, and Abhinav Gupta. "Robust adversarial reinforcement learning." ICML (2017).
>
> [2] Junru Sheng, Peng Zhai, Zhiyan Dong, Xiaoyang Kang, Chixiao Chen, and Lihua Zhang. Curriculum adversarial training for robust reinforcement learning. IJCNN (2022).
>
> [3] Jacob K Goeree, Charles A Holt, and Thomas R Palfrey. Quantal response equilibrium. In Quantal Response Equilibrium. Princeton University Press, 2016.

---

> > ### Comment · Reviewer_cFcL · 2023-11-22
> >
> > Thank you for all your detailed clarifications, which make this manuscript much clearer. I do not have any further questions, and I will keep my original score.

---

### Official Review · Reviewer_S4Ho · 2023-10-30

**Soundness:** 3 good
**Presentation:** 3 good
**Contribution:** 3 good
**Rating:** 8
**Confidence:** 4

**Summary:**

This paper builds on robust adversarial reinforcement learning (RARL) and add entropy regularization into the players' objectives. By adding regularization, the authors bound the rationality of the adversary (and protagonist) making the problem slightly easier to solve. Over training, the regularization is annealed, creating a curriculum, such that the ultimately trained protagonist is robust against a strong adversary. The authors show this approach outperforms several baselines empirically across a variety of tasks.

**Strengths:**

I found this paper generally easy to follow and well written. It proposes a simple modification to RARL that appears to greatly improve the empirical performance and make sense from a theoretical perspective as well.

**Weaknesses:**

Once the temperature had been split between the two players ($\alpha$ and $\beta$), I found the connection to QRE a bit tenuous. This is okay, but I think it would be better to present the QRE with one temperature and then state that you find better performance by using two.

I would have liked to see results over the course of training. Do you see monotonic improvement? How challenging is the saddle point problem (adversary vs protagonist) experimentally? Can you plot $(\alpha, \beta)$ over training?

**Questions:**

**I have upgraded my score after review of the authors' feedback**

- Note McKelvey and Palfrey defined the QRE along with a more specific QRE, called the limiting logit equilibrium, that is obtained by annealing the temperature from infinity to zero (homotopy approach).
- Equation 4: You describe this as a maximum entropy formulation, but this looks like an entropy regularized approach rather than selecting the equilibrium with maximum entropy (which is different).
- Irrational: I understand why you've chosen to pair "irrational" against "rational", but I think it's inaccurate. I think you mean "random". Note that a random policy is not necessarily irrational (e.g., random is Nash in rock-paper-scissors).
- Section 4.1: You say "In Markov games... QRE can be computed in closed form...". This is not true. If we could compute QREs in closed-form (at any temperature), we could compute nearly exact Nash equilibria in closed-form. I think you just mean that computing the denominator of equation 5 is difficult due to the integral. The integral becomes a sum with finite actions, but you still have to solve a fixed-point problem to compute a QRE.
- Why do you define a probability distribution over $\alpha$ instead of just controlling $\alpha$ point-wise?
- Equation 11: Do you minimize this for a fixed $\lambda$ and $\eta$?
- Do you have an easily-accesible reference for QREs with different temperature values per player (i.e., heterogeneous QREs)? I'm familiar with QREs, but never seen this and I couldn't track down the precise Goeree reference you cite. To my knowledge, this is a deviation from the QRE concept, but still makes for an interesting story as inspiration for your approach.

---

> ### Author Response · Authors · 2023-11-17
>
> > Once the temperature had been split between the two players ($\alpha$ and $\beta$), I found the connection to QRE a bit tenuous.
>
> We thank the reviewer for raising this point. We refer to Chapter 4 of [1], where heterogenous QRE theory is discussed. As our aim is to robustify the protagonist agent, we choose not to excessively bound the rationality of the protagonist so that it can reasonably solve the task at each stage and efficiently progress to the end of the curriculum, where the robustness benefits are maximized. Nevertheless, we find the suggestion of experimenting with homogenous QRE to be an interesting venture, and we have included it in a new ablation study in Appendix C.2 (Figure 18), where we compare homogeneous QRE against heterogeneous QRE in QARL. The results confirm that the use of heterogeneous QRE enables to achieve superior performance.
>
> > Results over the course of training. Do you see monotonic improvement? How challenging is the saddle point problem (adversary vs protagonist) experimentally? Can you plot (alpha, beta) over training?
>
> Despite QARL relying on entropy-regularization and, thus, not having guarantees of monotonic improvement, the empirical performance of the protagonist usually increases steadily thanks to the curriculum, which halts if the agent is no longer able to solve the task. For a typical example of what training looks like, we refer the reviewer to Figure 6a of Appendix A which plots the return of the protagonist agent and the agent temperatures at each iteration. Please note that the adversary return will simply be the negative of the protagonist return, and the protagonist temperature is tuned using the automatic temperature curriculum of SAC and is therefore independent of the curriculum.
>
> > Note McKelvey and Palfrey defined the QRE along with a more specific QRE, called the limiting logit equilibrium, that is obtained by annealing the temperature from infinity to zero (homotopy approach).
>
> Thank you, this is definitely correct. QARL utilises the same logic as the homotopy approach mentioned above, as described in Section 4.2 of our work.
>
> > Equation 4: You describe this as a maximum entropy formulation, but this looks like an entropy regularized approach rather than selecting the equilibrium with maximum entropy (which is different).
>
> We understand this concern. However, we have chosen to use these terms interchangeably in the same manner as [1,2,3]. We can change it to "entropy regularization" if the Reviewer will consider it important.
>
> > Irrational is inaccurate. I think you mean "random". Note that a random policy is not necessarily irrational (e.g., random is Nash in rock-paper-scissors).
>
> Thank you. In the revised version, we have corrected all mentions of 'irrational' behavior in our work with 'minimal rationality' or 'random'.
>
> > Section 4.1: You say "In Markov games... QRE can be computed in closed form...". This is not true...
>
> We agree that this statement is too broad, and we have updated it in the revised version.
>
> > Why do you define a probability distribution over instead of just controlling point-wise?
>
> We point the Reviewer to the ablation study in Figure 3 investigating other curriculum methods. **Point Curriculum** refers to a curriculum of point-wise values of $\alpha$ rather than a distribution. We find that our method (**Auto**) results in the greatest degree of robustness. The idea is that by sampling over several similar values of temperature, both the adversary and protagonist agent are better able to generalize their behaviour at each iteration of the algorithm, smoothing the progression of the curriculum.
>
> > Equation 11: Do you minimize this for a fixed $\lambda$ and $\eta$?
>
> The $\lambda$ and $\eta$ in Eq. 11 are the Lagrange multipliers and are not set by us; rather, the minimization described in (11) (which is equivalent to that described in (10)) is performed using a minimizer function from a scientific Python library (`scipy.optimize.minimize`), which automatically adjusts the Lagrange multipliers.
>
> > Easily-accesible reference for QREs with different temperature values per player (i.e., heterogeneous QREs)
>
> Thank you for bringing up this point. We point the Reviewer to Chapter 4 of [4], which describes the mathematical and implementation details of heterogeneous QRE and its deviations from homogeneous QRE.
>
> [1] Haarnoja, Tuomas, Aurick Zhou, Pieter Abbeel, and Sergey Levine. "Soft actor-critic: Off-policy maximum entropy deep reinforcement learning with a stochastic actor." ICML (2018).
>
> [2] Haarnoja, Tuomas, Haoran Tang, Pieter Abbeel, and Sergey Levine. "Reinforcement learning with deep energy-based policies." ICML (2017).
>
> [3] Eysenbach, Benjamin, and Sergey Levine. "Maximum entropy rl (provably) solves some robust rl problems." arXiv (2021).
>
> [4] Jacob K Goeree, Charles A Holt, and Thomas R Palfrey. Quantal response equilibrium. In Quantal Response Equilibrium. Princeton University Press, 2016.

---

> > ### Comment · Reviewer_S4Ho · 2023-11-20
> > **Follow-Up**
> >
> > Thank you for your explanations and for pointing me to additional experiments (Figure 6a and Appx C.2). Also, I was able to track down a PDF copy of [4], but with effort. I have listed an additional, more easily accessible reference for heterogeneous QREs that other readers might appreciate [1].
> >
> > Re the limiting logit equilibrium. I understand that you may have been inspired by the QRE homotopy, but I think suggesting "QARL utilises the same logic as the homotopy approach" would mislead the reader. The homotopy identified in their original work requires a sensitive annealing schedule. For instance, it sometime requires *increasing* the temperature for some time before decreasing it again. It is not the case that any annealing schedule will lead you to this unique equilibrium.
> >
> > Re optimizing the Lagrangian (11). The Lagrange multiplier technique reformulates a constrained optimization problem as a saddle-point (min-max) problem. How exactly are you using `scipy.optimize.minimize` to solve this?
> >
> > [1] Brian W. Rogers, Thomas R. Palfrey, Colin F. Camerer,
> > Heterogeneous quantal response equilibrium and cognitive hierarchies,
> > Journal of Economic Theory,
> > Volume 144, Issue 4,
> > 2009,
> > Pages 1440-1467,
> > ISSN 0022-0531,
> > https://doi.org/10.1016/j.jet.2008.11.010.
> > (https://www.sciencedirect.com/science/article/pii/S0022053108001713)

---

> > > ### Author Response · Authors · 2023-11-20
> > >
> > > We appreciate the Reviewer's feedback on our response.
> > >
> > > > I have listed an additional, more easily accessible reference for heterogeneous QREs that other readers might appreciate [1].
> > >
> > > Thank you for sharing this additional reference. We acknowledge that an open-source reference for heterogeneous QREs is important to the clarity of our paper, and we included it as an additional reference in Section 4 of our work.
> > >
> > > > "QARL utilises the same logic as the homotopy approach" would mislead the reader. The homotopy identified in their original work requires a sensitive annealing schedule. For instance, it sometime requires increasing the temperature for some time before decreasing it again. It is not the case that any annealing schedule will lead you to this unique equilibrium.
> > >
> > > We agree with the Reviewer. Our aim in mentioning the homotopy method was to highlight a conceptual similarity of our method with this well-established optimization technique. However, we have chosen to remove this point from our paper as we would rather not mislead a reader or distract from our method.
> > >
> > > > Re optimizing the Lagrangian (11). The Lagrange multiplier technique reformulates a constrained optimization problem as a saddle-point (min-max) problem. How exactly are you using scipy.optimize.minimize to solve this?
> > >
> > > Using `scipy.optimize.minimize` with the 'trust-constr' method solves a Lagrangian, as described here: https://docs.scipy.org/doc/scipy/reference/optimize.minimize-trustconstr.html#optimize-minimize-trustconstr.
> > >
> > > We explicitly show this Lagrangian in (11) and use the minimizer to solve it for us. We kindly refer the Reviewer to the function `update_temperature_distribution` in `code/src/teachers/self_paced_gamma_teacher_fixed_beta.py` in our supplied code to see how the minimizer is called during the curriculum. In practice, the objective function and constraints listed in (10) are supplied directly to `scipy.optimize.minimize`.

---

### Official Review · Reviewer_DPhM · 2023-10-31

**Soundness:** 3 good
**Presentation:** 4 excellent
**Contribution:** 3 good
**Rating:** 6
**Confidence:** 4

**Summary:**

In this paper, the authors propose a novel entropy regularization algorithm called Quantal Adversarial RL for adversarial reinforcement learning  which  modulates adversarial rationality to ease the complexity of solving saddle point optimization problem in robust adversarial RL.They connect entropy regularization in RL to bounded rationality and Quantal Response Equilibrium in game theory and show how temperature parameter in entropy regularization can control rationality and helps to train against a rational adversary. They provide extensive experiments showing QARL outperforms RARL and other baselines in several MuJoCo problems.

**Strengths:**

1. Connection between entropy regularization and bounded rationality is novel.
2. Proposes a novel constraint optimization problem to design curriculum for updating temperature coefficient that slowly changes an irrational adversary to a fully rational one.
3. Extensive empirical experiments to demonstrate the effective of their algorithm.

**Weaknesses:**

1. Relies on heuristic approach to design curriculum schedule with no theoretical guarantee on convergence behavior.
2. Certain parts of the paper and notations can be improved, see questions section.

**Questions:**

1. Where is policy $\pi$ used in the equation 6,7,8? What is variable $\mathcal P$ as the conditioning parameter?
2. Definition of $Q^\*$ and $\pi^\*$ is not very clear. Both $Q^\*$ and $\pi^\*$ depend on each other in eqn 6 and 8?
3. Is there any insight into why the curriculum approach helps optimization compared to direct adversarial training?
4. How well does the computation overhead of sampling different rationality level scale?

---

> ### Author Response · Authors · 2023-11-17
>
> > Relies on heuristic approach to design curriculum schedule with no theoretical guarantee on convergence behavior.
>
> We thank the Reviewer for raising this point. We acknowledge that as both adversarial and curriculum methods in deep RL are typically lacking in convergence guarantees, it is prohibitive to prove well-defined behaviour of our method in the theoretical sense. We do find that the heuristic curriculum parameters outlined in Table 4 of the Appendix are relatively easy to find during initial tuning of the algorithm and are broadly applicable to a wide range of environments, as evidenced by the improved empirical convergence properties of our algorithm compared to the tested baselines, as seen in Section 5.
>
> > Where is policy $\pi$ used in the equation 6,7,8?
>
> > What is variable P as the conditioning parameter?
>
> > Definition of $Q^*$ and $\pi^*$ and is not very clear. Both $Q^*$ and $\pi^*$ depend on each other in eqn 6 and 8?
>
> - In equations 6,7,8 the optimal soft value function $V^*$ and optimal soft action-value function $Q^*$ are those obtained when following the optimal temperature-conditioned policy $\pi^*$.
>
> - $\mathcal{P}$ is the transition probability density function, as defined in the Preliminaries Section 3.
>
> - $Q^*$ and $\pi^*$ are dependent on each other. $Q^*$ is the fixed point of the soft Bellman backup operator, and $\pi^*$ is the optimal soft policy that results from soft policy iteration to convergence. This formulation is typical for maximum entropy RL [1,2].
>
> > Is there any insight into why the curriculum approach helps optimization compared to direct adversarial training?
>
> Thank you for raising this point. It is known [3,4] that gradient-based algorithms in non-cooperative games are not always guaranteed to converge (e.g., due to bad initializations or a highly non-concave non-convex objective landscape). Entropy regularization is one method by which a difficult objective landscape can be made 'easier' to optimise over. In the single-player case, augmenting the objective with an entropy term relaxes the fluctations in a difficult objective landscape, facilitates the use of a higher learning rate, and reduces the number of local optima in the objective. These claims are studied qualitatively in [5], which sums up the benefits of MaxEnt RL as, 'The smoothing effect of entropy regularization, if decayed over the optimization procedure, is akin to techniques that start optimizing, easier, highly smoothed objectives and then progressively making them more complex'.
>
> > How well does the computation overhead of sampling different rationality level scale?
>
> The computational overhead of sampling temperatures scales linearly in the number of per-agent training episodes $N$ (Algorithm 1). In practice, this sampling contributes a trivial fraction of the computational overhead since we model the temperature using a closed-form distribution.
>
> The total curriculum process of each iteration (including performance estimation, updating the temperature distribution, and sampling) account for $\frac{1}{60}$ of the computational overhead on average. We would like to point that this can be also verified through our submitted codebase in the supplementary material.
>
> [1] Haarnoja, Tuomas, et al. "Reinforcement learning with deep energy-based policies." ICML (2017).
>
> [2] Haarnoja, Tuomas, et al. "Soft actor-critic: Off-policy maximum entropy deep reinforcement learning with a stochastic actor." ICML (2018).
>
> [3] Zhang, Kaiqing, et al. "On the stability and convergence of robust adversarial reinforcement learning: A case study on linear quadratic systems." NeurIPS (2020).
>
> [4] Zhang, Runyu, Zhaolin Ren, and Na Li. "Gradient play in stochastic games: stationary points, convergence, and sample complexity." arXiv (2021).
>
> [5] Ahmed, Zafarali, Nicolas Le Roux, Mohammad Norouzi, and Dale Schuurmans. "Understanding the impact of entropy on policy optimization." ICML (2019).

---

> > ### Comment · Reviewer_DPhM · 2023-11-21
> >
> > Thank you for your clarifications! I do not have any further questions, and would keep my score.

---

### Official Review · Reviewer_puRh · 2023-10-31

**Soundness:** 4 excellent
**Presentation:** 3 good
**Contribution:** 3 good
**Rating:** 6
**Confidence:** 3

**Summary:**

Robust Adversarial Reinforcement Learning (RARL) trains a protagonist against an adversary in a competitive zero-sum Markov game. However, in a high-dimensional control setting, finding the Nash equilibria faces complex saddle points. This method eases the complexity of the saddle point in optimization problems based on entropy regularization. They show the solution of an entropy-regularized problem corresponds to a Quantal Response Equilibrium, in which agents may be irrational with a certain probability. Based on this fact, this paper proposes an algorithm named Quantal Adversarial Reinforcement Learning (QARL). This algorithm first trains the protagonist against an irrational adversary and gradually increases the irrationality of the adversary until it is fully rational. This paper shows that QARL achieves stronger performance and robustness compared with other RARL algorithms in a wide range of reinforcement learning settings.

**Strengths:**

1. This paper is novel in the sense that it proposes a new method achieving stronger performance and robustness in adversarial reinforcement learning by gradually increasing the rationality of the adversary during training.

2. This paper draws an interesting connection between Quantal Response Equilibrium and an entropy-regularized problem and proposes a hyperparameter to smoothly control the rationality of the agent.

3. The experimental results are detailed and convincing. QARL outperforms other RARL algorithms in both performance and robustness. The experiments are conducted in a wide range of reinforcement learning settings including MuJoCo locomotion and navigation problems.

**Weaknesses:**

1. To train the protagonist by QARL, this paper requires that the rationality of the adversary can be controlled. This is a strong assumption, and its motivation is not well-justified. It would be better if this paper could show the performance of QARL even if the rationality of the adversary is not tuned but just increasing. Most importantly, this weakness limits the possible application of this algorithm to scenarios where a reliable simulator is available because the control over the rationality of the adversary may only be possible in a simulator.

2. Estimation of (12) consumes non-negligible online trajectories. The way to tune the rationality hyperparameter $\alpha$ is expensive.

**Questions:**

Have you tried any other heuristics for tuning $\alpha$? If so, could you briefly discuss them?

---

> ### Author Response · Authors · 2023-11-17
>
> > To train the protagonist by QARL, this paper requires that the rationality of the adversary can be controlled. This is a strong assumption ...
>
> We thank the Reviewer for pointing this out. We want to clarify that by leveraging the connection between entropy regularization and Quantal Response Equilibria, we show that we are able to control the rationality of the agent by simply changing the temperature parameter of the entropy regularization term. This is not different from the way regular Deep RL algorithms, e.g., SAC, tune the temperature during training, and it is independent to the environment and/or the used simulator. In fact, this can be done even when training online in real-world applications, as done in previous works [1].
>
> > The way to tune the rationality hyperparameter
>  is expensive.
>
> In practice, the $M$ rollouts described in Algorithm 1 required to estimate the expected performance of the protagonist agent result in negligible computational overhead when training in simulation. In one iteration of QARL (lines 2-23 of Algorithm 1), the majority of computation is used in the iterative training of the deep RL agents themselves, which is already present in the robust adversarial RL framework. Additionally, the minimization described in (10) is performed using a standard minimizer from Scipy (`scipy.optimize.minimize`) which results in $\frac{1}{60}$ of the computational overhead on average. We would like to point that this can be also verified through our submitted codebase in the supplementary material.
>
> > Have you tried any other heuristics for tuning?
>
> Thanks for raising this question. We acknowledge the importance of evaluating other approaches for the tuning of the temperature. In fact, we have already provided in the original submission an ablation study on different ways to tune the temperature, as it can be seen in Figure 3. We evaluate the obtained robustness on $2$ locomotion problems for $4$ different temperature tuning approaches:
> - **Auto:** Our automatic curriculum generation approach;
> - **Linear Curriculum:** Linearly annealing of the temperature, i.e., linear increase of rationality;
> - **Point Curriculum:** Using a scalar value to model the temperature instead of sampling it from a probability distribution;
> - **Reduced Sampling:** Sampling only one value at each iteration, i.e., $N=1$.
>
> We show that **Auto** outperforms the other approaches. This showcases that both the optimization problem (10) we propose and the use of a probability distribution to model the temperature are crucial in QARL.
>
> [1] Haarnoja, Tuomas, Aurick Zhou, Kristian Hartikainen, George Tucker, Sehoon Ha, Jie Tan, Vikash Kumar et al. "Soft actor-critic algorithms and applications." arXiv (2018).

---

### Author Response · Authors · 2023-11-17
**Comments about the revised version**

We thank the Reviewers for their valuable feedback. Based on the Reviewers' comments, we revised our paper and supplemental material to provide improved versions compared to our initial submission. To enhance clarity, we have highlighted the modifications in blue. In summary, the revised paper includes:

- We have added Figure 17 in Appendix C.2, which ablates the value of the performance threshold $\xi$ for the MuJoCo Cheetah - Run environment. We show that a relatively low value (as a fraction of the total achievable score for each environment) is the most reliable method for tuning $\xi$, as this progresses the curriculum under the condition that the protagonist has not catastrophically failed at solving the task at each iteration.

- We have added Figure 18 in Appendix C.2, which compares the progression of QARL with distinct protagonist and adversary temperatures (i.e., heterogeneous QRE), as done in all our experiments of the main paper, to homogeneous QRE, i.e., when both agents have the same temperature during training. This ablation supports our choice of bounding the rationality of the adversary with the curriculum while allowing the protagonist to follow its own temperature schedule.

- We have improved the writing incorporating the Reviewers' suggestions.


Please note that we have also reuploaded our code in the supplementary material to include the new experiment with homogeneous QARL as well as some general clarifying comments.

---

### Meta-Review · Area_Chair_Ahtj · 2023-12-03

**Metareview:**

This paper proposed a new framework for achieving adversarial robustness in RL. In contrast to the classic Robust Adversarial Reinforcement Learning (RARL) approach where the learner aims to learn the Nash equilibrium between the agent and adversary, which is known to be computationally difficult, this paper propose to instead solve for a equilibrium called the Quantal Response Equilibrium (QRE). The paper moves on to show that QRE can be approximated by performing annealing on a sequence of objective functions with entropy regularization. This insight is valuable in that it draws connection between RARL and entropy-regularized RL. Additionally, the paper conducts convincing experiments that demonstrate the superior performance of the proposed Quantal Adversarial RL (QARL) framework compared to RARL and recent baselines. Overall, this is a very solid paper and the reviewers are unanimous in accepting this paper.

**Justification For Why Not Higher Score:**

I believe orals belong to research with a larger breakthrough than what's presented in this paper. To the best of my knowledge, the question of RARL is still mainly of theoretical interests and not applied in any real-world problems.

**Justification For Why Not Lower Score:**

The technical contribution is strong and should be of interest to audiences within the RARL research community.

---

### Decision · Program_Chairs · 2024-01-16

Accept (spotlight)